# STalign: Alignment of spatial transcriptomics data using diffeomorphic metric mapping

Kalen Clifton [1,2,8], Manjari Anant[1,3,8], Gohta Aihara [1,2], Lyla Atta [1,2], Osagie K. Aimiuwu [4], Justus M. Kebschull[2,5], Michael I. Miller [2,5], Daniel Tward [6,7] ✉ & Jean Fan [1,2,5] ✉

Spatial transcriptomics (ST) technologies enable high throughput gene expression characterization within thin tissue sections. However, comparing spatial observations across sections, samples, and technologies remains challenging. To address this challenge, we develop STalign to align ST datasets in a manner that accounts for partially matched tissue sections and other local non-linear distortions using diffeomorphic metric mapping. We apply STalign to align ST datasets within and across technologies as well as to align ST datasets to a 3D common coordinate framework. We show that STalign achieves high gene expression and cell-type correspondence across matched spatial locations that is significantly improved over landmark-based affine alignments. Applying STalign to align ST datasets of the mouse brain to the 3D common coordinate framework from the Allen Brain Atlas, we highlight how STalign can be used to lift over brain region annotations and enable the interrogation of compositional heterogeneity across anatomical structures. STalign is available as an open-source Python toolkit at https://github.com/JEFworks-Lab/STalign and as Supplementary Software with additional documentation and tutorials available at https://jef.works/STalign.

Spatial transcriptomics (ST) technologies have enabled high-throughput, quantitative profiling of gene expression within individual cells and small groups of cells in fixed, thin tissue sections. Comparative analysis of ST datasets at matched spatial locations across tissues, individuals, and samples provides the opportunity to interrogate spatial gene expression and cell-type compositional variation in the context of health and disease. Such comparative analysis is complicated by technical challenges such as in sample collection, where the experimental process may induce tissue rotations, tears, and other structural distortions. Other challenges include biological variation such as natural inter-individual tissue structural differences. In order to

reliably characterize spatial molecular differences between ST datasets along comparative axes of interest, it is integral to control for potentially confounding tissue structural variation by spatially aligning these tissue structures across ST datasets.

Considering the recent development of such ST technologies, options for spatially aligning across ST datasets are still limited. Previous computational methods have focused on spatial alignment of ST datasets for which each dataset is assayed using the same pixel-resolution ST technology with only a few hundred to a few thousand spatial measurements[1,2]. These methods face challenges in scaling to larger, single-cell resolution ST datasets with tens to hundreds of

[1]Center for Computational Biology, Whiting School of Engineering, Johns Hopkins University, Baltimore, MD, USA. [2]Department of Biomedical Engineering, Johns Hopkins University, Baltimore, MD, USA. [3]Department of Neuroscience, Johns Hopkins University, Baltimore, MD, USA. [4]University of North Carolina at Chapel Hill, Chapel Hill, NC, USA. [5]Kavli Neuroscience Discovery Institute, The Johns Hopkins University, Baltimore, MD, USA. [6]Department of Computational Medicine, University of California Los Angeles, Los Angeles, CA, USA. [7]Department of Neurology, University of California Los Angeles, Los Angeles, CA, USA. [8]These authors contributed equally: Kalen Clifton, Manjari Anant. ✉e-mail: dtward@mednet.ucla.edu; jeanfan@jhu.edu

thousands of spatial measurements. Further, spatial alignment of datasets across different ST technologies remains challenging. Other alignment methods are limited to rigid, affine transformation such as based on landmarks[3] and cannot accommodate non-linear distortions. To address these challenges, we present an approach called STalign that builds on recent developments in Large Deformation Diffeomorphic Metric Mapping[4,5] (LDDMM) to align ST datasets using image varifolds. STalign is amenable to data from single-cell resolution ST technologies as well as data from multi-cellular pixel-resolution ST technologies for which a corresponding registered single-cell resolution image such as a histology image is available. STalign is further able to accommodate alignment in both 2D and 3D coordinate systems. STalign is available as an open-source Python toolkit at https://github.com/JEFworks-Lab/STalign and as Supplementary Software with additional documentation and tutorials available at https://jef.works/STalign.

## Results

### Overview of Method

To align two ST datasets, STalign solves a mapping that minimizes the dissimilarity between a source and a target ST dataset subject to regularization penalties. Within single-cell resolution ST technologies, both the source and target ST datasets are represented as cellular positions $(x^{\rho_S}, y^{\rho_S})$ and $(x^{\rho_T}, y^{\rho_T})$ respectively (Fig. 1a, Supplementary Note 1). Solving the mapping with respect to single cells has quadratic complexity and is computationally intractable, so STalign applies a rasterization approach to reduce computational time (Fig. 1b, Supplementary Note 2). Briefly, STalign models the positions of single cells

as a marginal space measure $\rho$ within the varifold measure framework[6]. STalign then convolves the space measure $\rho$ with Gaussian kernels $k$ to obtain the smooth, rasterized function $I(x,y) = [k^{\frac{1}{2}} * \rho](x,y)$. Finally, STalign samples from the continuous $I(x,y)$ to get a discrete image of a specified size with a specified pixel resolution. For ST technologies or atlases that are not single-cell resolution, instead of rasterizing cellular positions to generate an image, we can utilize other registered standard image types for alignment including red-green-blue images of hematoxylin and eosin (H&E) stained tissue.

To solve for a mapping that minimizes the dissimilarity between source and target images $I^S$ and $I^T$, STalign utilizes the LDDMM framework (Fig. 1c). Using LDDMM to identify a diffeomorphic solution allows us to have a smooth, continuous, invertible transformation which permits mapping back and forth from the rasterized image and original cell positions while respecting the biological constraints such that cell neighbor relationships stay relatively the same[7]. The mapping $\phi^{A,v}$ is constructed from two transformations, an affine transformation $A$ and a diffeomorphism $\varphi_1^v$ such that $\phi^{A,v}(x) = A\varphi_1^v(x)$, where $\varphi_1^v$ is generated by integrating a time-varying velocity field $v_t$ over time and $A$ acts on $\varphi_1^v(x)$ through matrix vector multiplication in homogeneous coordinates (Supplementary Note 3). STalign focuses on solving for a mapping that minimizes the dissimilarity between the source and target images $I^S$ and $I^T$ rather than minimizing the dissimilarity between the source and target space measures because, while approximately equivalent, the former can be calculated more efficiently (Supplementary Note 4).

The optimal $\phi^{A,v}$ is computed by minimizing an objective function that is the sum of a regularization term, $R(v)$, and a matching term,

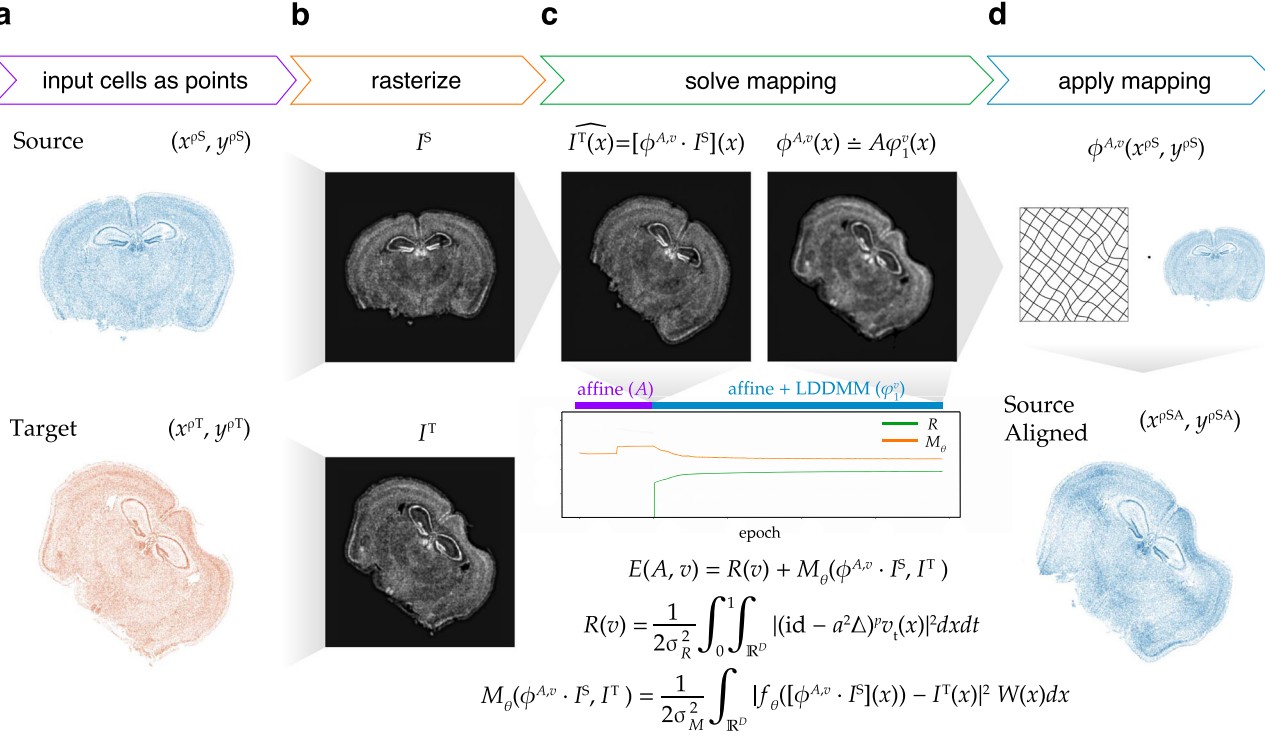

**Fig. 1 | Overview of STalign on ST data from a single-cell resolution technology. a** STalign takes as input a source (blue) and target (orange) ST dataset as x- and y-coordinates of cellular positions, $(x^{\rho_S}, y^{\rho_S})$ and $(x^{\rho_T}, y^{\rho_T})$. **b** Source and target coordinates are then rasterized into images $I^S$ and $I^T$. **c** To align $I^S$ and $I^T$, STalign solves for the mapping $\phi^{A,v}$ that when applied to $I^S$ estimates $I^T$ such that $I^T(x) = [\phi^{A,v} \cdot I^S](x)$. Gradient descent is used to solve affine transformation $A$ and large deformation diffeomorphic metric mapping (LDDMM) $\varphi_1^v$ that compose $\phi^{A,v}$ such that $\phi^{A,v}(x) = A\varphi_1^v(x)$. The objective function minimized includes a regularization term $R(v)$ to penalize non-smooth solutions (where id is an identity matrix, $a$

is a spatial smoothness constant, $\Delta$ is the Laplacian, $p$ is a power, and $v_t$ is a time varying velocity field) and a matching term $M_\theta(\phi^{A,v} \cdot I^S, I^T)$ that minimizes the dissimilarity between the transformed source image and the target image while accounting for tissue and technical artifacts with $W(x)$ and $f_\theta$, respectively. Balance between regularization and matching accuracy can be tuned with the parameters $\sigma^2_R$ and $\sigma^2_M$. Components of the objective function decrease over epochs with transforms at different stages of the diffeomorphism. **d** Once $\phi^{A,v}$ is solved, visualized as a deformation field, the mapping is applied to the coordinates of the source to obtain the coordinates for the aligned source, $(x^{\rho_{SA}}, y^{\rho_{SA}})$.

$M_\theta(\phi^{A,v} \cdot I^S, I^T)$. The relative weights of the regularization term and matching term can be tuned with $\sigma^2_R$ and $\sigma^2_M$. The regularization term controls spatial smoothness. In this term, we optimize over $v_t, t \in [0,1]$ noting that if $v_t$ is constricted to being a smooth function, the $\varphi^v_1$ constructed from $v_t$ is guaranteed to be diffeomorphic. The matching term incorporates a Gaussian mixture model $W(x)$ to estimate matching, background, and artifact components of the image to account for missing tissue such as due to partial tissue matches or tears. Additionally, the matching term contains an image contrast function $f_\theta$ to account for differences due to variations in cell density and/or imaging modalities. To solve all parameters in each term a steepest gradient descent is performed over a user-specified number of epochs. Once $\phi^{A,v}$ is computed, STalign applies this computed transformation to the source's original cell positions $(x^{\rho_S}, y^{\rho_S})$ to generate aligned source coordinates $(x^{\rho_{SA}}, y^{\rho_{SA}})$ (Fig. 1d, Supplementary Note 5).

## STalign enables alignment of single-cell resolution ST datasets within technologies

As a proof of concept, we first applied STalign to align two single-cell resolution ST datasets from the same technology. Specifically, we aligned, in a pairwise manner at matched locations, ST data from 9 full coronal slices of the adult mouse brain representing 3 biological replicates spanning 3 different locations with respect to bregma assayed by MERFISH (Methods). Inherent local spatial dissimilarities between slices, due to biological variability and further exacerbated by technical variation as well as tears and distortions sustained in the data acquisition process, render affine transformations such as rotations and translations often insufficient for alignment.

To evaluate the performance of STalign, we first evaluated the spatial proximity of manually identified structural landmarks between the source and target ST datasets, expecting the landmarks to be closer together after alignment. We manually placed 12 to 13 landmarks that could be reproducibly identified (Supplementary Fig 1, Supplementary Table 1). To establish a supervised affine transformation for comparison with STalign, we solved for the affine transformation that minimized the error between these landmarks using least squares. We then compared the positions of the corresponding landmarks after both the supervised affine alignment and STalign alignment using root-mean-square error (RMSE). When the supervised affine transformations were used for alignment, RMSE was 202 +/- 17.1 μm, 170 +/- 3.47 μm, and 266 +/- 6.65 μm for biological replicates of each slice location respectively. When STalign based on an LDDMM transformation model was used for alignment, RMSE was 113 +/- 10.5 μm, 169 +/- 4.53 μm, and 175 +/- 5.47 μm for biological replicates of each slice location respectively. STalign was thus able to consistently reduce the RMSE between landmarks after alignment compared to an affine transformation, suggestive of higher alignment accuracy.

Given the ambiguity of where landmarks may be manually reproducibly placed and their inability to evaluate alignment performance for the entire ST dataset, we next took advantage of the available gene expression measurements to further evaluate the performance of STalign. Because of the highly prototypic spatial organization of the brain, we expect high gene expression correspondence across matched spatial locations after alignment. We focused our evaluation on one pair of ST datasets of coronal slices from matched locations (Methods). We visually confirm that alignment results in a high degree of spatial gene expression correspondence (Fig. 2a, Supplementary Fig. 2a). To further quantify this spatial gene expression correspondence, we evaluated the gene expression magnitudes at matched spatial locations across the aligned ST datasets. Specifically, we aggregated cells into pixels in a 200 μm grid to accommodate the differing numbers of cells across slices and then quantified gene expression magnitude correspondence at spatially matched 200 μm pixels using cosine similarity (Fig. 2b, c,

Supplementary Fig. 2b). For a good alignment, we would expect a high cosine similarity approaching 1, particularly for spatially patterned genes. To identify such spatially patterned genes, we applied MERINGUE[8] to identify 457 genes with highly significant spatial auto-correlation (Methods). For these genes, we observe a high spatial correspondence after alignment as captured by the high median cosine similarity of 0.73. In contrast, for the remaining 192 non-spatially patterned genes, we visually confirm as well as quantify the general lack of spatial correspondence (Fig. 2d–f, Supplementary Fig. 3a, b). We note that these non-spatially patterned genes are enriched in negative control blanks (57%), which do not encode any specific gene but instead represent noise such that we would not expect spatial correspondence even after alignment. Further, we observe a low median cosine similarity of 0.21 across non-spatially patterned genes ($n = 192$) that is significantly lower than for spatially patterned genes ($n = 457$) (Wilcoxon rank-sum test $p$-value < 2.2e-16).

We next compare the alignment achieved with STalign to the alignment from a supervised affine transformation based on our previously manually placed landmarks (Supplementary Fig. 4a, Methods). We visually confirm that a supervised affine alignment results in a lower degree of spatial gene expression correspondence than alignment by STalign (Supplementary Fig. 4b). We again evaluate the performance of the supervised affine transformation using a pixel-based cosine similarity quantification (Supplementary Fig. 4c). We find that for spatially patterned genes, the cosine similarity is consistently higher with a mean difference of 0.09 for the alignment by STalign compared to supervised affine (Supplementary Fig. 4d). In contrast, for non-spatially patterned genes, the cosine similarity is more comparable with a mean difference of 0.02 for the alignment by STalign compared to supervised affine (Supplementary Fig. 4e). This greater improvement in spatial gene expression correspondence for the alignment achieved with STalign compared to supervised affine transformation for spatially patterned genes suggests that modeling non-linearity in alignment with approaches like STalign can achieve a higher alignment accuracy compared to linear alignment approaches.

## STalign enables alignment of ST datasets across technologies

Many technologies for spatially resolved transcriptomic profiling are available, varying in experimental throughput and spatial resolution[9]. We thus applied STalign to align two ST datasets from two such different ST technologies. Specifically, we applied STalign to align the previously analyzed single-cell resolution ST dataset of a full coronal slice of the adult mouse brain assayed by MERFISH to a multi-cellular pixel resolution ST dataset of an analogous hemi-brain slice assayed by Visium (Fig. 3a). As such, in addition to being from different ST technologies, these two ST datasets further represent partially matched tissue sections. Because of this partial matching, we incorporated manually placed landmarks to initialize the alignment as well as further help steer our gradient descent towards an appropriate solution (Supplementary Note 6). For the MERFISH data, we rasterized the cell positions to generate the source image. For the Visium dataset, the target image for alignment was a red-green-blue single-cell resolution H&E staining image obtained from the same tissue section as the gene expression data (Methods).

To evaluate the performance of this alignment, we again take advantage of the available gene expression measurements. Due to partially matched tissue sections, we restricted downstream comparisons to tissue regions STalign assessed with a matching probability > 0.85 (Methods). We again visually confirm that the spatial alignment results in a high spatial gene expression correspondence albeit at differing resolutions across the two technologies (Fig. 3b, Supplementary Fig. 5a). To further quantify this spatial gene expression correspondence, we evaluated the gene expression magnitudes at matched spatial locations across the aligned tissue sections for the 415 genes with non-zero expression in both ST datasets. We evaluated

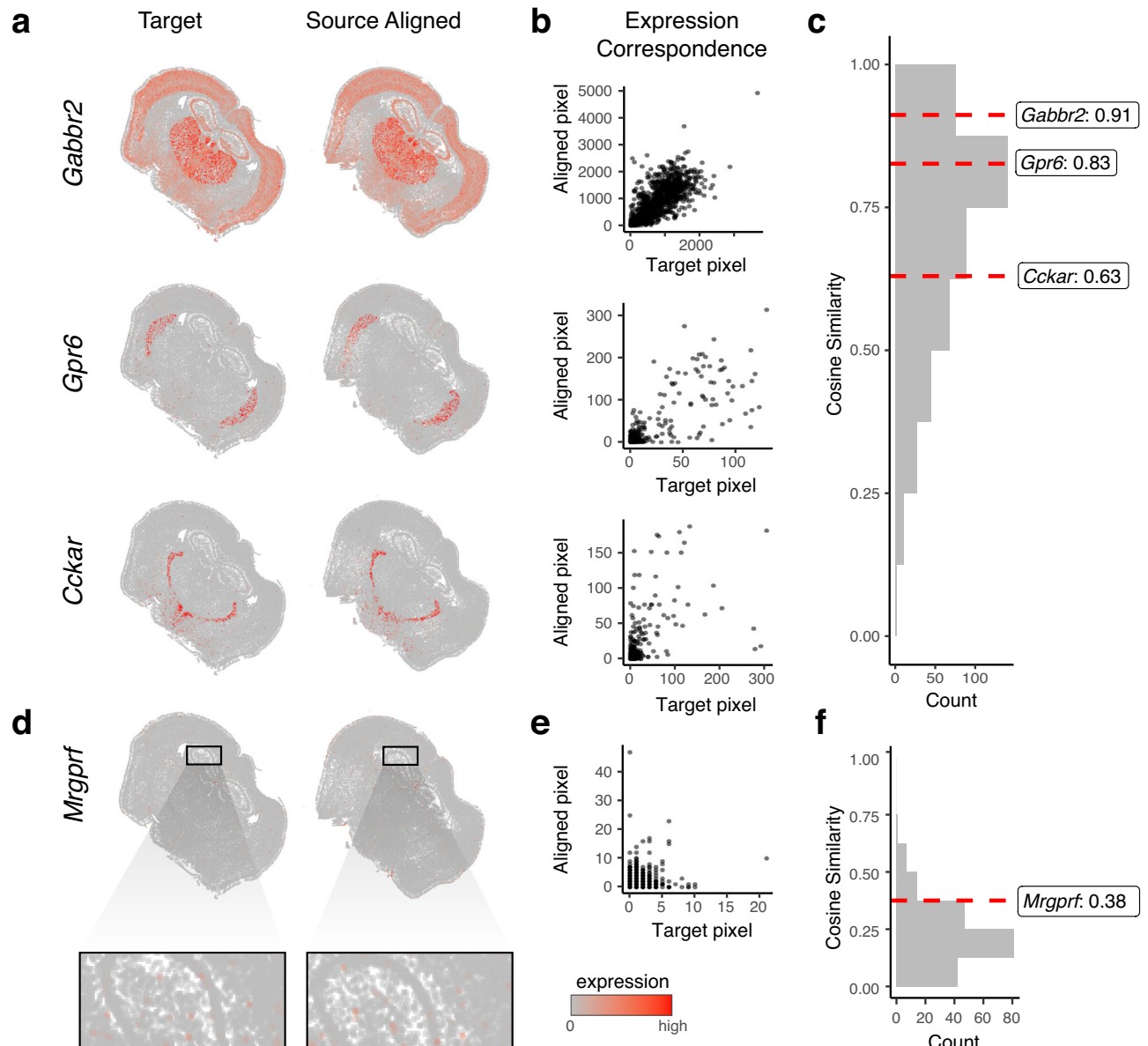

**Fig. 2 | Evaluation of STalign based on spatial gene expression correspondence.** **a** Correspondence of gene expression spatial organization between the target and aligned source for select spatially patterned genes. **b** Transcript counts in the target compared to the aligned source at matched pixels for select genes: *Gabbr2*, *Gpr6*, and *Cckar*. **c** Distribution of cosine similarities between transcript counts in target versus aligned source at matched pixels for 457 spatially patterned genes with select genes marked. **d** Spatial pattern of expression for a select non-spatially patterned gene in the target and aligned source (inset displays cells at higher magnification). **e** Counts for the target versus aligned source at matched pixels for a select non-spatially patterned gene, *Mrgprf*. **f** Distribution of cosine similarities between counts in target compared to the aligned source at matched pixels for 192 non-spatially patterned genes.

these genes for spatial autocorrelation on the Visium data to identify 227 spatially patterned genes and 188 non-spatially patterned genes (Methods). Due to the resolution differences between the two technologies, to ensure appropriate comparisons, we used the positions of the Visium spots to aggregate MERFISH cells into matched resolution pseudospots. Likewise, to control for detection efficiency differences between the two technologies, we performed the same counts-per-million normalization on the Visium spot gene expression measurements and the aggregated MERFISH pseudospots gene expression measurements (Fig. 3c, Supplementary Fig. 5b). We again evaluated gene expression correspondence at spatially matched spots using cosine similarity and observed a median cosine similarity of 0.55 across spatially patterned genes (Fig. 3d) and a median cosine similarity of 0.06 across non-spatially patterned genes (Supplementary Fig. 6a–c). We note that this gene expression correspondence after spatial

alignment is lower than what was previously observed within technologies most likely due to variation in detection efficiency across technologies in addition to variation in tissue preservation rather than poor spatial alignment. While MERFISH detects targeted genes at high sensitivity, Visium enables untargeted transcriptome-wide profiling though sensitivity for individual genes may be lower[9]. Likewise, while the MERFISH dataset was generated with fresh, frozen tissue, the Visium dataset was generated with FFPE-preserved tissue. Still, we anticipate that while sensitivity to specific genes may vary across technologies and with different tissue preservation techniques, the underlying cell-types should be consistent.

Therefore, we sought to evaluate the performance of our alignment based on cell-type spatial correspondence. To identify putative cell-types, we performed transcriptional clustering analysis on the single-cell resolution MERFISH data (Supplementary Fig. 7a) and

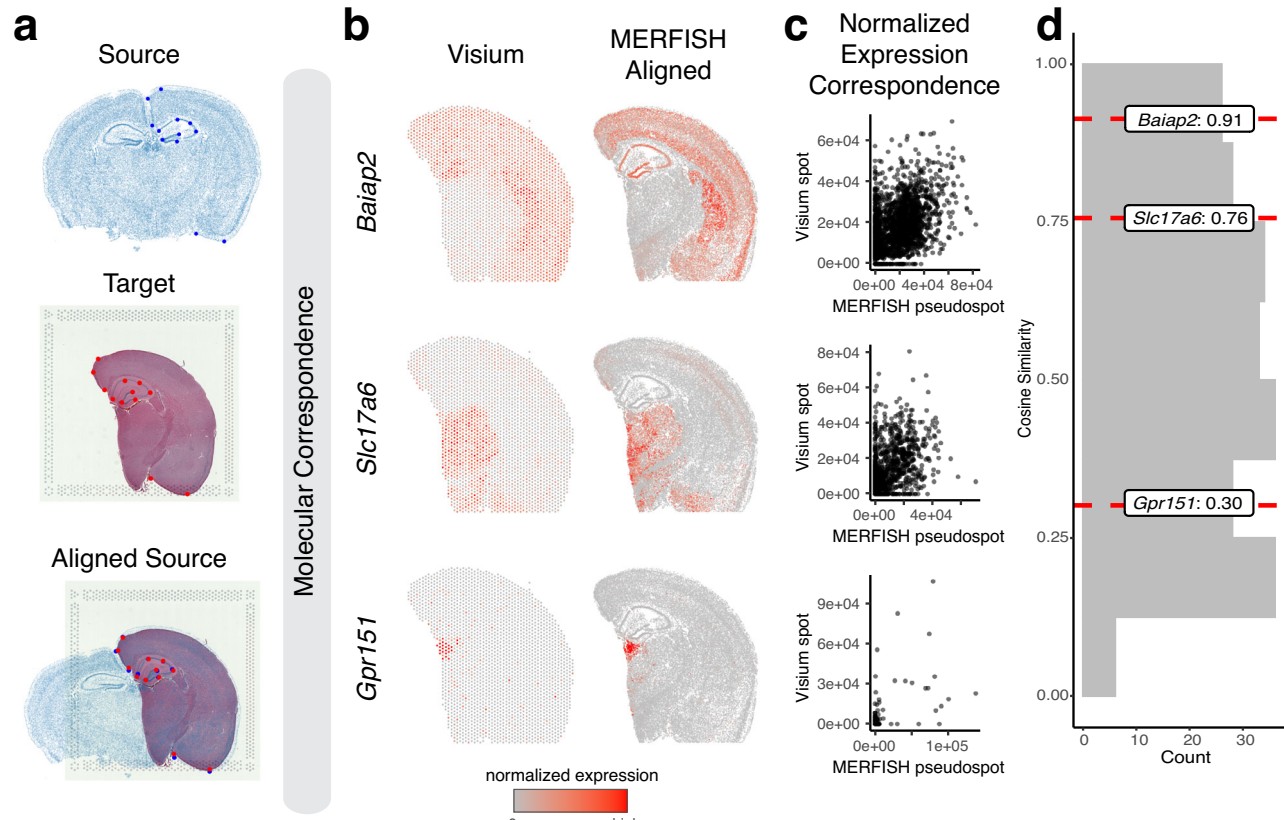

**Fig. 3 | Application and evaluation of STalign on spatial transcriptomics data from different ST technologies based on normalized spatial gene expression correspondence. a** Overview of STalign on ST data from different ST technologies. Single-cell resolution ST is used as the source, with the initial image being produced from the x- and y-coordinates of each cell's position (top). For the multi-cellular resolution ST technologies, the corresponding single-cell resolution histological image is used as the target (middle). STalign aligns the source to the target (bottom). The manually placed landmarks that were utilized to improve alignment for these partially matched tissues are marked in blue for the source and red for the target. **b** Correspondence of gene expression spatial organization between the Visium target and aligned MERFISH source for select spatially patterned genes. **c** Normalized gene expression in the Visium target compared to the aligned MERFISH source at matched spots and pseudospots respectively for select spatially patterned genes: *Baiap2*, *Slc17a6*, and *Gpr151*. **d** Distribution of cosine similarities between normalized gene expression in the Visium target versus aligned MERFISH source at matched spots and pseudospots for 227 spatially patterned genes detected by both ST technologies with select genes marked.

deconvolution analysis[10] on the multi-cellular pixel-resolution Visium data (Fig. 4a, Methods). We matched cell-types based on transcriptional similarity between cell clusters and deconvolved cell-types (Supplementary Fig. 7b). Indeed, we visually observe high spatial correspondence across matched cell-types (Fig. 4a, b). We evaluated the proportional correspondence of cell-types at aligned spot and pseudospot spatial locations by cosine similarity and observed a high median cosine similarity of 0.75 across cell-types (Fig. 4c, d). As such, STalign achieves high cell-type spatial correspondence across aligned ST datasets, suggestive of high alignment accuracy.

## STalign enables alignment of ST datasets to a 3D common coordinate framework

Tissues are inherently 3-dimensional (3D), and tissue sections are subject to distortions in 3D as well as 2D. As such, a more precise spatial alignment of 2D tissue sections must accommodate this 3D distortion. The underlying mathematical framework for STalign is amenable to alignment in 2D as well as 3D (Supplementary Note 7). We thus applied STalign to align ST datasets to a 3D common coordinate framework (CCF). Specifically, we applied STalign to align 9 ST datasets of the adult mouse brain assayed by MERFISH to a 3D 50 μm resolution grayscale volume of the adult mouse brain CCF established by the Allen Brain Atlas[11] (Methods, Fig. 5a).

We note that when aligning 2D ST datasets to a 3D CCF, there may not exist a perfect 2D hyperplane in the 3D space in which the

alignment falls. For example, when MERFISH Slice 2 Replicate 2 was aligned to the Allen Brain Atlas, the RMSE with respect to the best-fit 2D plane is 1.255 μm (Methods, Fig. 5b); this non-zero RMSE highlights the slice has deformities that extend out of the 2D hyperplane. Additionally, the angle of the best-fit 2D plane with respect to the y-z plane of the Allen Brain Atlas is 9.07 degrees; this non-zero angle demonstrates that the slice does not align with a strictly coronal section. As such, our 2D-3D alignment accommodates global deformations (i.e. 2D plane with respect to the 3D atlas) as well as local distortions (i.e. deformations in and out of a 2D plane) using the same underlying LDDMM framework as 2D alignments.

In the construction of the Allen Brain Atlas CCF, brain regions were delineated based on several features like cellular architecture, differential gene expression, and functional properties via modalities such as histological stains, in situ hybridization, and connectivity experiments to generate a set of reference brain region annotations[11]. By aligning to this CCF, we can lift over these annotations to each cell (Fig. 5c, Supplementary Fig. 8a), enabling further evaluation of variations of gene expression and cell-type composition within and across these annotated brain regions.

To assess the performance of our atlas alignment and lift-over annotations, we first confirmed the enrichment of genes within certain brain regions. Numerous previous studies have shown that some brain regions can be demarcated based on the expression of particular genes[12,13]. We use these characteristic gene expression patterns to

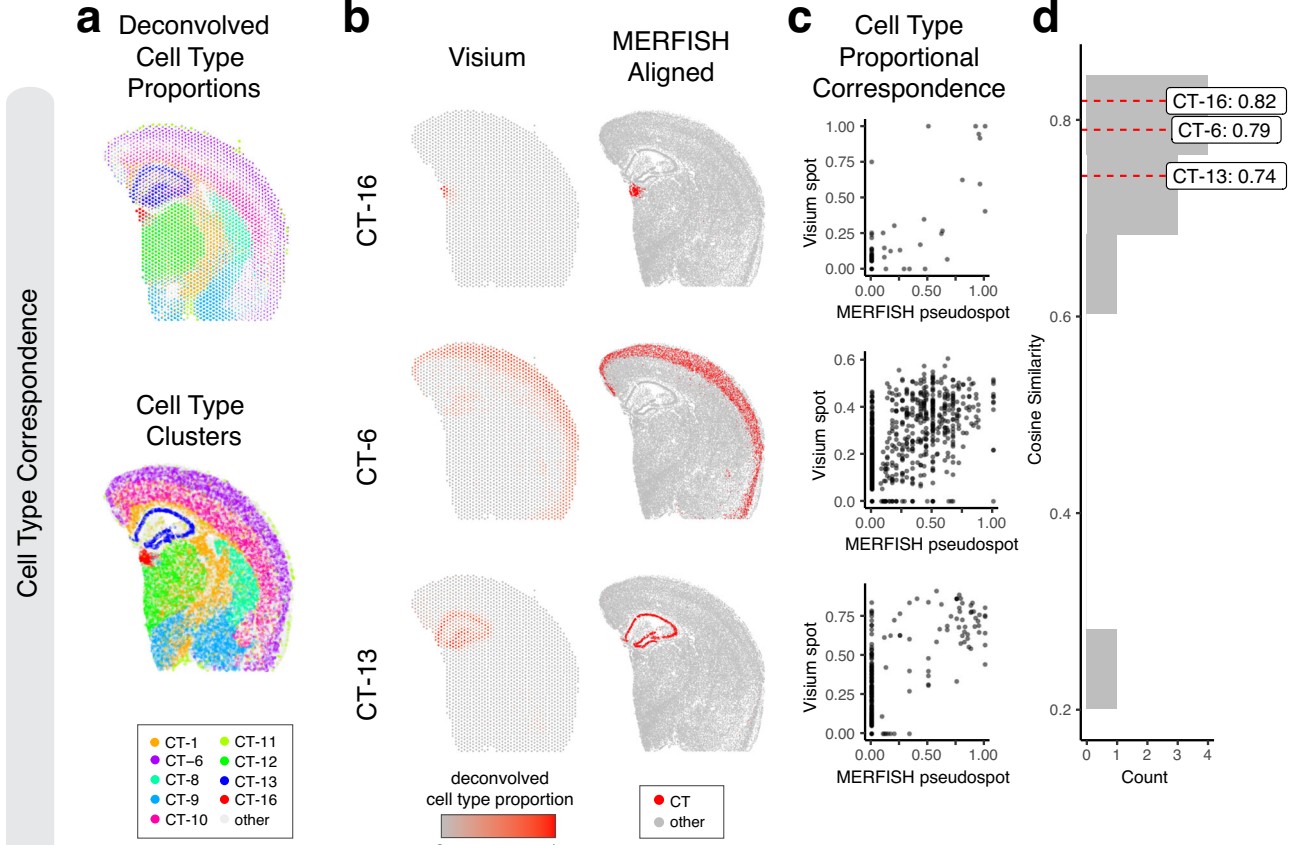

**Fig. 4 | Evaluation of STalign on ST data from technologies at different resolutions based on cell-type correspondence. a** Transcriptionally matched cell-types (CT) from deconvolution analysis of spot-resolution Visium data (top) and clustering analysis of spatially aligned single-cell-resolution MERFISH data (bottom). **b** Cell type correspondence between the Visium target and aligned MERFISH source with select cell-types shown. **c** Correspondence of cell-type proportion between the Visium target and aligned MERFISH source at matched spots and pseudospots respectively for select cell-types. **d** Distribution of cosine similarities between cell-type proportions in the Visium target and aligned MERFISH source at matched spots and pseudospots respectively for all matched cell-types with cell-types marked.

evaluate whether the brain regions lifted over from the Allen Brain Atlas CCF by STalign indeed contain expression of known marker genes. Consistent with previous studies, we found *Grm2* to be visually primarily enriched in the dentate gyrus brain region[14], *Sstr2* to be enriched in cerebral cortical layers 5 and 6 brain region[15], and *Gpr161* to be enriched in the CA1 brain region[16] (Fig. 5d), which was consistent across replicates (Supplementary Fig. 8b).

Next, we took a more agnostic approach to assess the performance of our atlas alignment and lift-over annotations by evaluating the consistency of cell-type compositional heterogeneity within brain structures across replicates. To identify cell-types, we perform unified transcriptional clustering analysis on these 9 ST datasets to identify transcriptionally distinct cell clusters and annotate them as cell-types based on known differentially expressed marker genes (Methods, Fig. 5e, f, Supplementary Fig. 9a). Many brain regions are known to have a characteristic cell type distribution[17–19]. Consistent with previous studies[20], we observed cell-types to be spatially and compositionally variable across brain regions (Fig. 5c, e). We visually confirmed that this spatial and compositional variability is consistent across replicates (Supplementary Fig. 8a, Supplementary Fig. 9b). To further quantify this consistency, for each brain region, we evaluated whether its cell-type composition was more similar between replicates than compared to a randomly demarcated brain region of matched size (Methods). For an accurate atlas alignment, we would expect the lift-over brain region annotations to be more similar in cell-type

composition across replicates, particularly for brain structures with distinct cell-type compositions, as compared to random brain regions of matched size. Indeed, we found that cell-type composition was generally more similar between replicates than compared to a random brain region of matched size (Fig. 5g). For example, for one replicate pair, we found that in 93% of evaluated brain structures (131/141), the cell-type composition was significantly more similar (paired t-test *p*-value = 6.805e-121) between replicates than compared to a random brain region of matched size. For the 7% (10/141) of brain regions that were less similar across replicates, we found that the number of cells in these brain regions was significantly fewer (Wilcoxon rank-sum test *p*-value = 0.002) than in other brain regions (Supplementary Fig. 10a). Notably, 60% of these brain regions had a minimum width of under 50 μm, including both compact and long, thin structures (Supplementary Fig. 10b), highlighting potential limitations with respect to alignment accuracy of such structures at this given resolution of alignment.

Finally, we also sought to assess the performance of our atlas alignment and lift-over annotations by evaluating cell-type compositions within and beyond annotated brain region boundaries (Methods). Specifically, we compare the entropy of each brain region based on the region's cell-type composition to entropy if the boundaries of these regions were expanded. Again, due to the characteristic cell-type distributions within brain regions in which one or a few cell-types predominate, we would expect accurate lift-

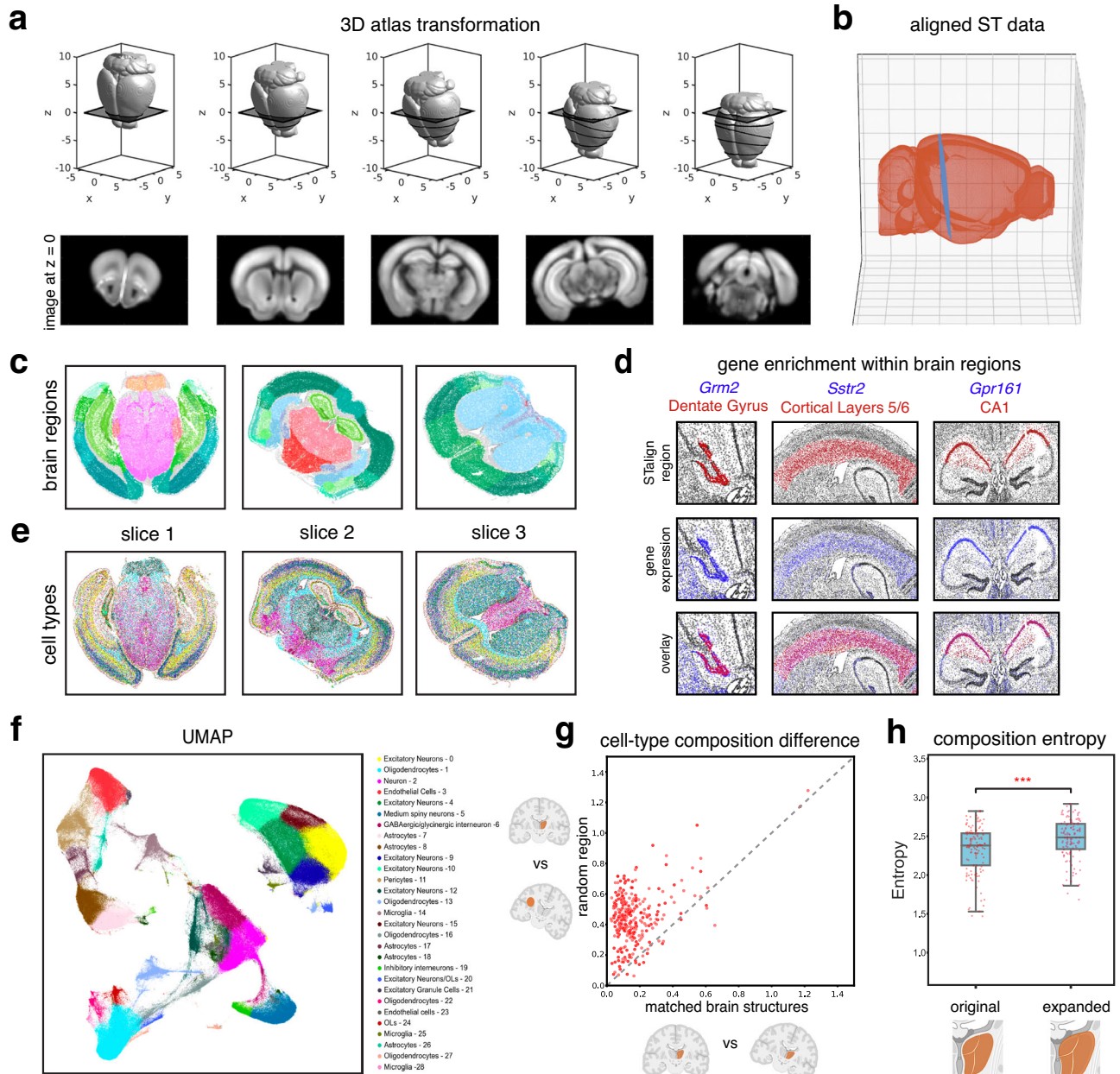

**Fig. 5 | Evaluation of 3D-2D alignment using STalign. a** Transformation of 3D CCF atlas to align to ST data at z = 0. **b** Aligned ST data (MERFISH Slice 1 Replicate 1) plotted in 3D Allen Brain Atlas coordinates. **c** Lift-over brain regions from aligning to the Allen Brain Atlas CCF with STalign. Cells are colored according to the colors assigned to brain regions by the Allen Brain Atlas. **d** Brain regions (top) labeled by STalign with expression of expected genes (middle) and overlay (bottom). **e** Spatial location of cell types on MERFISH brain slices. Legend for cell types provided in panel **f**. **f** UMAP embedding of different cell types defined by differential gene expression and Leiden clustering. **g** Cell-type composition difference between paired brain regions from two MERFISH replicates. The x axis represents cell-type composition difference within matched brain structures annotated by STalign across replicates and the y axis represents cell-type composition difference between STalign-annotated regions and size-matched random brain regions (n = 431 representing up to 148 shared brain regions across three replicate pairs). **h** Significant difference between distribution of cell-type composition entropy for brain regions labeled by STalign versus regions expanded by 100 nearest neighbors (center line, median; box limits, upper and lower quartiles; whiskers, 1.5x inter-quartile range; all data points shown, n = 148 brain regions; ***, statistically significant p-value = 8.6e-18). Created with Biorender.com.

over brain region annotations to exhibit entropies that are comparatively lower than if the boundaries of these regions were expanded, as more cell-types would be incorporated into the region and entropy would increase. We, therefore, expanded the brain structures lifted over by STalign by 100 nearest neighbors (NN), or approximately 100 μm, and evaluated the change in entropy. We performed the same analysis on randomly demarcated brain regions of matched size, which were expanded by 100 NN to account for increases in entropy due to the incorporation of more cells. We found

that the entropies for the original brain region annotations lifted over by STalign were significantly lower (paired t-test p-value = 8.6e-18) than for the expanded regions (Fig. 5h). In contrast, the entropies for randomly demarcated brain regions were not significantly lower (paired t-test p-value = 0.12) for the expanded regions (Supplementary Fig 11). Taken together, these results demonstrate that STalign can align ST datasets to a 3D CCF to consistently lift over atlas annotations that recapitulate the unique gene expression and cell-type composition within brain regions.

## STalign applicable to diverse tissues profiled by diverse ST technologies

STalign relies on variations in cell densities that generally form visible structures that can be used for alignment. As we have shown, alignment across samples and animals is possible for tissues with highly prototypic structures such as the brain. We further highlight the applicability of STalign to the diverse ST technologies by applying STalign to achieve structural correspondence for partially matched slices of the adult mouse brain assayed by two additional different single-cell resolution ST technologies, Xenium[21] and STARmap PLUS[22] (Methods, Fig. 6a–c).

For other tissues with substantially more inter-sample and inter-animal variation, alignment across serial sections is still achievable. For example, for serial sections of the developing human heart assayed at single-cell resolution with in situ sequencing (ISS)[23], we can apply STalign to achieve structural correspondence (Methods, Fig. 6d–f). Likewise, even for cancer tissues, which are highly non-prototypic in structure, there may still be sufficient structural consistency across serial sections to enable alignment. As such, we have applied STalign to align single-cell resolution ST datasets arising from partially matched serial sections of the same breast cancer sample assayed by Xenium (Methods, Fig. 6g–i). Likewise, we have applied STalign to align a single-cell ST dataset assayed by Xenium to a corresponding H&E image of the same tissue section (Methods, Fig. 6j–l). We visually observe a high degree of spatial correspondence and overlap of structural features after alignment, highlighting STalign's applicability to diverse tissues.

## Discussion

Alignment of ST datasets is a prerequisite step to enable comparisons across samples, subjects, and technologies. Alignment can also enable the pooling of measurements across biological replicates to construct consensus ST profiles[1] as well as enable 3D reconstruction by serial registration[24]. Here, we presented STalign, which builds on advancements in LDDMM, to perform alignment of ST datasets in a pairwise manner within ST technologies, across ST technologies, as well as to a 3D common coordinate system. We have shown that STalign achieves high accuracy based on the spatial proximity of manually identified shared landmarks as well as gene expression and cell-type correspondence at matched spatial locations after alignment. We note that based on these metrics, STalign outperforms affine transformations alone, highlighting the utility of local, non-linear transformations in alignment. STalign can further accommodate partially matched tissue sections, where one tissue section may be a fraction of another. We further apply STalign to align ST datasets to a 3D CCF to enable automated lift-over of CCF annotations such as brain regions in a scalable manner. We confirm that lift-over brain region annotations identify cells that express expected genes for a variety of brain regions. We also show that brain region annotations lifted over by STalign exhibit consistent cell-type compositions across replicates and within boundaries compared to random brain regions matched in size.

We anticipate that future applications of STalign to ST data from serial sections or conditionally matched tissues across different ST technologies will enable cross-technology comparisons as well as cross-technology integration through spatial alignment. In particular, applying STalign to align ST data of structurally similar tissues from the same condition across different ST technology platforms may allow us to better interrogate platform-specific differences and strengths. Given that different ST technologies currently generally prioritize either resolution or genome-wide capabilities, applying STalign to align serial sections assayed by different ST technologies may allow us to leverage each technology's unique strengths to characterize matched spatial locations. With atlasing efforts like The Human BioMolecular Atlas Program and others producing 3D CCFs[25], the application of STalign to align ST data to such CCFs to enable automated lift-over of

atlas structural annotations will facilitate standardization and unification of biological insights regarding annotated structures. Likewise, STalign complements gene-expression-based approaches for sample alignment[26] by focusing on the real space rather than a higher-order transcriptomic manifold. We further anticipate future applications of STalign to ST data from structurally matched tissues in case-control settings assayed by the same ST technologies will enhance the throughput for yielding meaningful comparisons regarding gene expression and cell-type distributions in space as evidenced by recent applications of ST technologies to characterize spatially-resolved age-related[27] and injury-related[28] gene expression variation.

As ST technologies continue to evolve, we anticipate STalign will continue to be applicable due to our use of rasterization to convert the positions of single cells into an image with a specified resolution. The runtime of each iteration of the STalign alignment algorithm scales with respect to the number of pixels in this image. For most evaluated datasets, we find that STalign is generally able to converge onto an optimal alignment within a few minutes to a few hours, depending on the number of pixels, the number of iterations, and other system variables (Methods, Supplementary Table 2). Whereas other alignment algorithms generally scale in memory and runtime with the number of spatially resolved measurements (spots or cells)[1,2], which will likely make them computationally untenable as ST technologies evolve to increase the number of spatially resolved measurements that can be assayed. Overall, we anticipate that the ability for users to choose the rasterization resolution, and therefore the number of pixels in the rasterized image, will allow STalign to maintain its utility for larger datasets.

Still, among the limitations of STalign with respect to ST data, it is currently applicable to only ST datasets with single-cell resolution or those accompanied with a registered single-cell resolution histology image from the same assayed tissue section, which may not be available to all non-single-cell resolution ST technologies. STalign further relies on the representative nature of cell segmentations in ST data to reflect underlying tissue structures. As such, limitations in cell segmentations that render the derived cell density no longer representative of the profiled tissue structure could present challenges for alignment with STalign.

Further, as STalign is based on an LDDMM transformation model for alignment, it inherits the same limitations. As LDDMM relies on optimization using gradient descent, the resulting alignment solution may converge on local minima. Strategies to guide the optimization away from potential local minima may be applied in the future. Likewise, the more different the source and targets for alignment, particularly for partially matching sections, the more important the initialization will be for this optimization. As we have shown, landmark points may be used to guide the initialization of an orientation and scaling for alignment. In addition, LDDMM enforces an inverse consistency constraint such that every observation in the target must have some correspondence in the source in a manner that cannot accommodate holes or other topological differences in the tissue through the deformation only[7]. As such, when performing alignments, we advise choosing the more complete tissue section as the source because our Gaussian mixture modeling for accommodating partially matched tissues and other artifacts applies to the target image intensity only.

Still, alignment accuracy at the resolution of single cells is limited by the fact that there is generally no one-to-one correspondence between cells across samples, particularly for complex tissues. As such, accuracy can typically only be expected to be achieved up to a mesoscopic scale at which it is reasonable to define cell density[29]. As we have shown, this presents challenges, particularly in aligning thin structures. While STalign currently uses an isotropic (Gaussian) kernel to estimate cell densities, future work considering non-isotropic kernels may improve accuracy for these thin structures. However, generally, our choice of kernel will inherently bias our alignment towards

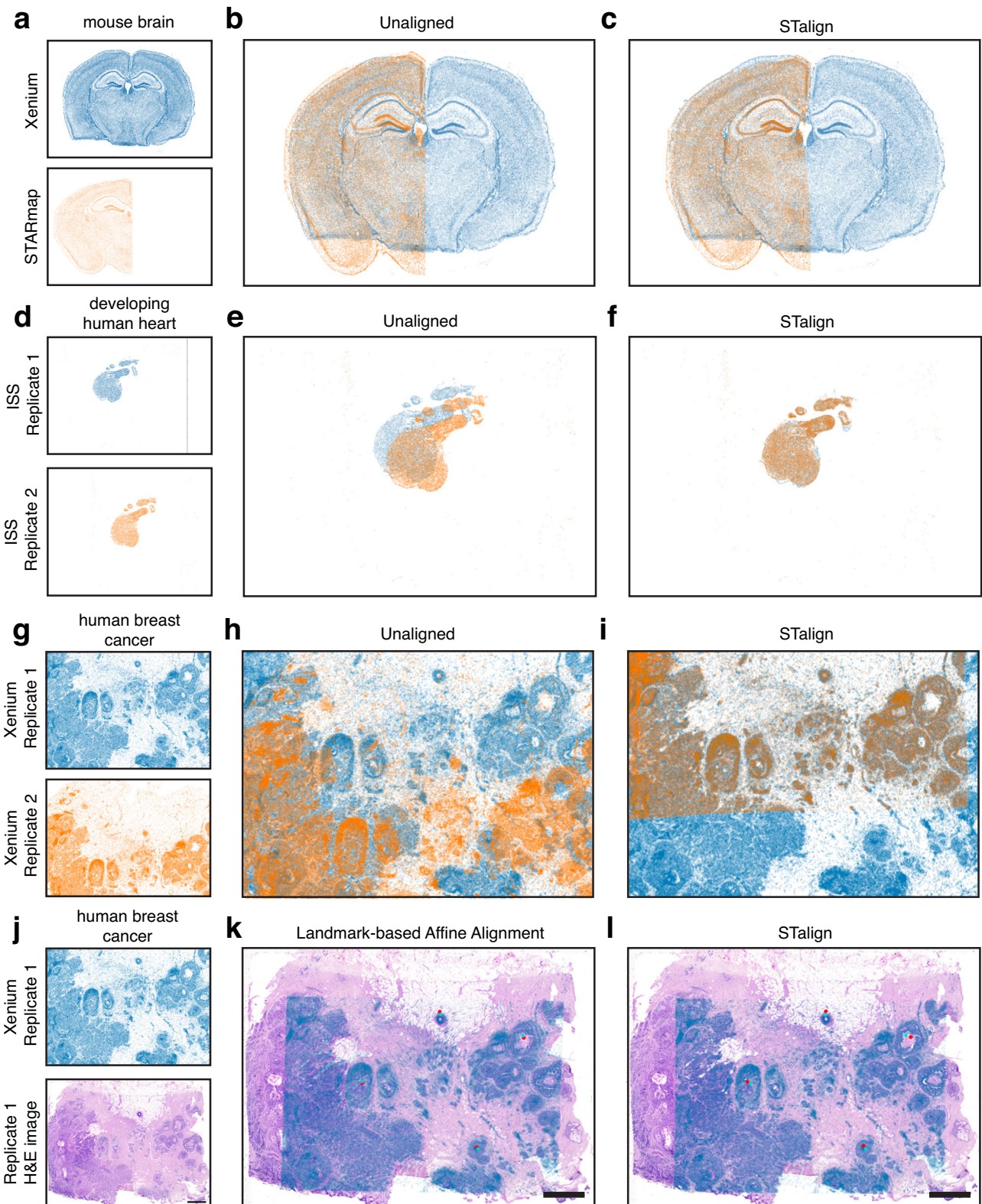

**Fig. 6 | Application of STalign to ST data of diverse tissues. a** Two coronal slices of the adult mouse brain assayed by two different single-cell resolution ST technologies, Xenium and STARmap PLUS **b** Overlay of cellular positions before alignment. **c** Overlay of cellular positions after alignment with STalign. **d** Two single-cell resolution datasets, Replicate 1 and Replicate 2, from serial sections of the developing human heart assayed by ISS and visualized as x- and y-coordinates of cellular positions. **e** Overlay of cellular positions before alignment. **f** Overlay of cellular positions after alignment with STalign. **g** Two single-cell resolution ST datasets, Replicate 1 and Replicate 2, from partially matched, serial breast cancer sections assayed by Xenium and visualized as x- and y-coordinates of cellular positions. **h** Overlay of cellular positions before alignment. **i** Overlay of cellular positions after alignment with STalign. **j** Replicate 1 of the Xenium single-resolution breast cancer dataset with a corresponding H&E image from the same tissue section. **k** Overlay of cellular positions and H&E image based on affine transformation by minimizing distances between manually placed landmarks, shown as points in red and turquoise. **l** Overlay of cellular positions and H&E image after alignment with STalign. Scale bar 1 mm.

accuracy at a certain structural scale. Likewise, although we focused here on aligning based on cell densities, STalign and the underlying LDDMM framework can also be applied to align using cellular features such as gene expression magnitude, reduced dimensional representations of gene expression such as via principal components, or cell-type annotations, which may improve the accuracy of alignment for regions with homogenous cell density but heterogeneous gene expression and cell-type composition. However, integration of such features in the alignment process necessitates orthogonal means of performance evaluation beyond the correspondences in gene expression magnitude and cell-type proportions that we have used here. By aligning based on cell densities, we do not require shared gene expression quantifications or unified cell-type annotations, potentially enhancing flexibility and providing opportunities for integrating across other data modalities for which spatially resolved single cell resolution information is available such as other spatial omics data in the future.

Overall, we anticipate that moving forward STalign will help provide a mathematical framework for ST data alignment to enable integration and downstream analyses requiring spatial structural alignment to potentially reveal new insights regarding molecular and cell-type compositional differences between different tissue structures and across various physiological axes.

## Methods

### Datasets

Nine MERFISH datasets consisting of 734,696 cells and 483 total genes, across 9 brain slices (3 replicates of 3 coronal sections from matched locations with respect to bregma) were obtained from the Vizgen website for *MERFISH Mouse Brain Receptor Map data release* (https://info.vizgen.com/mouse-brain-map).

A Visium dataset of an FFPE-preserved adult mouse brain were obtained from the 10X Datasets website for *Spatial Gene Expression Dataset by Space Ranger 1.3.0* (https://www.10xgenomics.com/resources/datasets/adult-mouse-brain-ffpe-1-standard-1-3-0).

A Xenium dataset (In Situ Replicate 1) of a fresh frozen mouse brain coronal section was obtained from the 10X Datasets website for *Mouse Brain Dataset Explorer* (https://www.10xgenomics.com/products/xenium-in-situ/mouse-brain-dataset-explorer).

STARMAP Plus data (well11_spatial.csv) of coronal slices of the adult mouse brain was downloaded from the Broad Single Cell Portal (https://singlecell.broadinstitute.org/single_cell/study/SCP1830/spatial-atlas-of-molecular-[…]pes-and-aav-accessibility-across-the-whole-mouse-brain).

Developing heart data for samples CN73_E1 and CN73_E2 were downloaded from the Human Developmental Cell Atlas (https://hdca-sweden.scilifelab.se/a-study-on-human-heart-development/) via ST_heart_all_detected_nuclei.RData (https://github.com/MickanAsp/Developmental_heart).

Two Xenium datasets (In Situ Replicate 1 and In Situ Replicate 2) of a single breast cancer FFPE tissue block were obtained from the 10X Datasets website for *High-resolution mapping of the breast cancer tumor microenvironment using integrated single cell, spatial and* in situ *analysis of FFPE tissue* (https://www.10xgenomics.com/products/xenium-in-situ/preview-dataset-human-breast).

The CCF and brain region annotations were obtained from the Allen Brain Atlas API (https://help.brain-map.org/display/mouseconnectivity/API). The 50 μm resolution 3D adult mouse brain CCF used for alignment was ara_nissl_50 can be downloaded directly (http://download.alleninstitute.org/informatics-archive/current-release/mouse_ccf/ara_nissl/ara_nissl_50.nrrd). The brain region annotations that correspond to the 50um resolution CCF were ccf_2017/annotation_50 downloaded can be directly (https://download.alleninstitute.org/informatics-archive/current-release/mouse_ccf/annotation/ccf_2017/annotation_50.nrrd).

### Application of STalign

To align MERFISH datasets, we applied STalign in a pairwise manner across replicates for sections from matched locations with respect to bregma, rasterized at a 50 μm resolution, and iterated over 1000 epochs, with the following changes to default parameters (sigmaM: 0.2).

To align a MERFISH dataset to a Visium dataset, we applied STalign with MERFISH Slice 2 Replicate 3, rasterized at a 50 μm resolution, as the source and the red-green-blue high-resolution Visium hematoxylin and eosin (H&E) staining image as the target. We utilized the landmark points stored in Merfish_S2_R3_points.npy and tissue_hires_image_points.npy as inputs pointsI and pointsJ. We iterated for 200 epochs with the following changes to default parameters (sigmaP: 0.2, sigmaM: 0.18, sigmaB: 0.18, sigmaA: 0.18, diffeo_start: 100, epL: 5e-11, epT: 5e-4, epV:5e1).

To align MERFISH to the Allen CCF, we applied STalign using the 3D reconstructed Nissl image from the Allen CCF atlas as a source, and each of our 9 MERFISH images as a target. The Nissl CCF is a 50 μm resolution grayscale volume that was constructed by registering 2D Nissl images to a 3D anatomical template. The template to which this Nissl dataset was registered was constructed by averaging high-resolution customized serial two-photon tomography scans of 1675 brains[11].

To align Xenium and STARmap datasets of mouse brain coronal sections, we applied STalign with Xenium In Situ Replicate 1, rasterized at 30 μm resolution, as the source and STARmap well 11, rasterized at 30 μm resolution, as the target. Prior to rasterization, STARmap cell centroid positions were scaled by 1/5 such that the overlay of unaligned sections showed both Xenium and STARmap cells positions at a similar scale. We iterated for 1000 epochs with the following changes to default parameters (sigmaM:1.5, sigmaB:1.0, sigmaA:1.5, epV: 100, muB: black).

To align serial developing heart sections, we applied STalign with sample CN73_E1 as the source and CN73_E2 as the target, both rasterized at 100 μm resolution. We iterated for 1000 epochs with the following changes to default parameters (diffeo_start:100, a: 250, sigmaB:0.1, epV: 1000, muB: black).

To align Xenium datasets, we applied STalign with Xenium Breast Cancer Replicate 1 as the source and with Xenium Breast Cancer Replicate 2 as the target, rasterized at 30 μm resolution. We placed a set of 3 manually chosen landmark points to compute an initial affine transformation. We iterated for 200 epochs with the following changes to default parameters (sigmaM:1.5, sigmaB:1.0, sigmaA:1.5, epV: 100).

To align Xenium to H&E, we applied STalign with Xenium Breast Cancer Replicate 1, rasterized at 30 μm resolution, as the target and the corresponding red-green-blue H&E image from the same tissue as the source. We placed a set of 3 manually chosen landmark points to compute an initial affine transformation. We iterated for 2000 epochs with the following changes to default parameters (sigmaM:0.15, sigmaB:0.10, sigmaA:0.11, epV: 10, muB: black, muA: white) where muB and muA initializes the mixture model for the background and artifact components as corresponding to black and white colors respectively in the target image.

### Expression based performance evaluation for STalign-based alignment of single-cell resolution ST datasets within technologies

To evaluate the performance of STalign on aligning datasets from the same technologies based on expression correspondence, we focused on the alignment of Slice 2 Replicate 3 and Slice 2 Replicate 2 from the MERFISH datasets, with the former as the source and the latter as the target.

A grid was created to partition all cells into 200 μm square pixels. For each 200 μm pixel, the gene expression of cells in the pixel was

summed for the aligned source and for the target to get gene expression at 200 μm resolution.

MERINGUE (v1.0) was applied to calculate Moran's I on the 200 μm resolution summed gene expression of the target. Genes with an adjusted $p$-value < 0.05 were identified as significantly spatially patterned genes and genes with an adjusted $p$-value >= 0.05 were identified as non-significantly spatially patterned genes.

For each gene, the cosine similarity was calculated between the 200 μm resolution summed gene expression counts in the aligned source and the 200 μm resolution summed gene expression counts in the target across pixels. A Wilcoxon rank sum test was used to compare the distributions of cosine similarities for spatially patterned and non-significantly spatially patterned genes.

### Comparison to supervised affine alignment of single-cell resolution ST datasets within technologies

In addition to alignment by STalign, we performed supervised affine alignment of Slice 2 Replicate 3 and Slice 2 Replicate 2 from the MERFISH datasets, with the former as the source and the latter as the target. We manually placed 13 landmarks in the source and target that could be reproducibly identified (Supplementary Fig 1, Supplementary Table 1) using our script point_annotator.py. We solved for the affine transformation that minimized the error between these landmarks using least squares and applied the affine transformation to the cell positions of the source. With the supervised affine aligned source and target, we repeated the expression-based performance evaluation described in section "Expression-based performance evaluation for STalign-based alignment of single-cell resolution ST datasets within technologies".

### Evaluation alignment across technologies

**Expression based performance.** Given that the MERFISH tissue section is larger than the Visium, we considered the aligned region to be limited to the MERFISH tissue that had a matching probability > 0.85 based on the posterior probability of pixels belonging to the matched class in the Gaussian mixture modeling, with the 0.85 threshold being manually chosen based on visual inspection. We restricted the set of cells in the MERFISH dataset to only those in this aligned region for downstream evaluation.

To aggregate the cells in the aligned MERFISH dataset into pseudospots that match with the Visium spots, we calculated the distances between the positions of the MERFISH cells and the positions of the Visium spot centroids. Cells were classified as within the pseudospot that corresponds to the Visium spot if the distance of the cell to the Visium centroid was less than the Visium spot radius. The Visium spot radius information was obtained by multiplying the spot_diameter_fullres by the tissue_hires_scalef in the Visium scalefactors_json.json file and dividing by 2. For each pseudospot, the gene expression of all cells within the pseudospot was summed.

For gene expression correspondence analysis, we restricted to the 415 genes that had at least one copy in both the MERFISH and Visium datasets and that were detected in more than one spot in the Visium dataset.

MERINGUE (v1.0) was applied to calculate Moran's I on the Visium counts-per-million (CPM) normalized counts. Genes with an adjusted $p$-value < 0.05 were identified as significantly spatially patterned genes and genes with an adjusted $p$-value >= 0.05 were identified as non-significantly spatially patterned genes.

CPM normalization and log10 transformation with a pseudocount of 1 were applied on the gene expression of the MERFISH pseudospots and Visium spots. For each gene, the cosine similarity was calculated between the normalized and log-transformed gene expression magnitudes across matched MERFISH pseudospots and Visium spots.

**Cell-type correspondence performance.** To identify cell-types in the Visium data, we applied STdeconvolve (v1.6.0) on a corpus of 838 genes after filtering out lowly expressed genes (<100 copies), genes present in <5% of spots and genes present in > 95% of spots and restricting to significantly over-dispersed with alpha =1e-16 to obtain a corpus <1000 genes, resulting in 16 deconvolved cell-types.

To identify cell-types in the aligned MERFISH data, PCA was performed on the CPM normalized cell by gene matrix. Louvain clustering was performed on a neighborhood graph of cells using the top 30 PCs and 90 nearest neighbors to identify 16 transcriptionally distinct clusters of cells.

To match deconvolved cell-types and single-cell clusters, we used the deconvolved cell-type-specific transcriptomic profiles from STdeconvolve and averaged the transcriptional profiles per cluster from single-cell clustering. We restricted to the 257 shared genes, CPM normalized, and correlated the resulting normalized transcriptional profiles using Spearman correlation. We considered a Visium deconvolved cell-type and MERFISH single-cell cluster as a match if they had transcriptional similarity > 0.5.

For each matched cell-type, we evaluated spatial compositional correspondence using cosine similarity of the cell-types proportional representation across matched MERFISH pseudospots and Visium spots.

### Evaluation of 2D to 3D CCF alignment

**Quantification of local and global distortions.** To evaluate the local deformation of the 2D MERFISH Slice 2 Replicate 2 when aligned to the 3D Allen Brain Atlas CCF, the STalign function LDDMM_to_slice() was run with the following parameters: nt=4, niter=2000, sigmaA = 2, sigmaB = 2, sigmaM = 2, muA = [3,3,3], muB = [0,0,0]. Then, the least squares fit method was used to evaluate the most representative plane for the transformed slice, and the root mean square error was calculated. The angle between the most representative plane and the y-z plane, which runs through the medio-lateral and dorso-ventral axis of the Allen Brain Atlas, was computed.

**Unified transcriptional clustering analysis and cell-type annotation.** All MERFISH datasets were combined. Transcriptional clustering analysis and cell type annotation was performed using the SCANPY package[30] [version 1.9.1]. Data were normalized to counts per million (scanpy: normalize_total) and log transformed (scanpy: log1p). PCA (scanpy: pca) was computed on the cell by gene matrix. A neighborhood graph of cells using the top 10 PCs and 10 nearest neighbors was created (scanpy: neighbors), and Leiden clustering was performed on this graph (scanpy: leiden) to identify 29 clusters. Differentially expressed genes were extracted from each cluster (scanpy: rank_genes_groups), and cell-types were annotated based on marker genes in each cluster.

**Annotated brain region composition analysis.** To generate randomly demarcated brain regions, a random number generator (random.randint) defined the x, y coordinate of the center of the random region, and the random region was composed of the N closest points to the center, where N is the number of cells in the brain region. A slice/replicate with random regions was constructed for all slice/replicates with STalign annotated regions, and the number of cells (N) were the same for STalign and randomly demarcated regions.

To compare cell-type compositions, each region was represented by a cell-type vector, which was composed by the proportion of each cell type in the region (29 x 1 vector). We calculate the Euclidean distance between cell-type vectors of the same region across replicates in Slice 2 using the regions annotated by STalign. The Euclidean distance was also found across replicates in Slice 2 using randomly demarcated brain regions and STalign brain

regions. 431 data points for each group were used, comparing replicate 1 to replicate 2, replicate 2 to replicate 3, and replicate 3 to replicate 1 for up to 148 shared brain regions across replicate pairs. Further characterization was performed for one representative MERFISH replicate pair (Slice 2 Replicates 2 and 3) to characterize differences in Euclidean distance for the 141 shared brain regions. The Euclidean distances of cell type vectors between STalign brain regions replicate and between randomly demarcated brain regions were compared using a paired t-test.

To evaluate annotated brain region boundaries, brain regions were expanded using k-nearest neighbors (k = 100) using the ball tree algorithm for each region and each replicate in Slice 2 (sklearn.-neighbors.NearestNeighbors). The procedure was conducted for STalign annotated brain regions and randomly demarcated brain regions. Shannon's entropy was evaluated for STalign annotated and randomly demarcated brain regions that were expanded by 100 nearest neighbors. Paired t-tests were used to compute p-values between original and expanded brain regions for STalign and random groups. Effect size was computed as a difference in the means of the compared distributions. PP plots were used to visualize normality, and we used a Gaussian fit with R > 0.8 and a variance ratio less than 4 to confirm normality and equal variances. 431 data points for each group were used.

To evaluate regions that had a greater Euclidean distance between two STalign regions compared to random versus STalign regions, we calculated the number of cells and Shannon's entropy of each region and tested for significance using a Wilcoxon Rank Sum test due to the small sample size. Shannon's entropy was calculated using the formula $\sum p(x) * \log(p(x))$ where p(x) is the probability of picking cell-type x from the given brain region (scipy.special.entr).

### Statistics & reproducibility

No data were excluded from the statistical analyses. No statistical method was used to predetermine sample size. The experiments were not randomized. The Investigators were not blinded to allocation during experiments and outcome assessment.

**Evaluation of alignment within technologies.** Statistical analyses were performed by the R package stats version 4.2.1. p-value was determined by Wilcoxon rank-sum test, and p-value < 0.05 is considered statistically significant.

**Evaluation of 2D-3D alignment.** Statistical analyses were performed by the SciPy Python package, v1.11. p-value was determined by paired t-test (Fig. 5h, Supplementary Fig. 11) and Wilcoxon rank-sum test (Supplementary Fig. 10), and p-value < 0.05 is considered statistically significant.

### Implementation

The implementation of STalign as STalign.py (Supplementary Software) uses the parameters and default values provided in Table 1.

The PyTorch framework was used for automatic gradient calculations. Based on the PyTorch backend, STalign supports parallelization across multiple cores or on GPUs. Derivatives (covectors) are converted to gradient vectors[5,31] for natural gradient descent[32].

For improved robustness, STalign allows users to input pairs of corresponding points in the source and target images. These points can be used to initialize the affine transformation A through least squares to steer our gradient-based solution toward an appropriate local minimum in this challenging nonconvex optimization problem as well as be added to the objective function to drive the optimization problem itself. Landmark-based optimization in the LDDMM framework has been studied extensively[33]. A script point_annotator.py is provided to assist with the interactive placement of these points (Supplementary Software).

**Table 1 | Parameters and default values of functions STalign.rasterize() and STalign.LDDMM()**

| Symbol | Explanation | Default |
|---|---|---|
| dx | Width of rasterization kernel | 30 µm |
| σM | Weight on image-matching functional | 1.0 |
| σR | Weight on regularization matching functional | 5.00E + 05 |
| σP | Weight on landmark matching functional | 2.00E + 01 |
| σA | Variance of artifact component for Gaussian Mixture Modeling | 5 |
| σB | Variance of background component for Gaussian Mixture Modeling | 2 |
| a | Smoothness scale of diffeomorphism | 500.0 µm |
| p | Power of Laplacian for regularization | 2 |
| niter | Number of iterations of gradient descent | 5000 |
| diffeo_start | Iteration to start optimizing $v_t$ for coarse-to-fine | 0 |
| nt | Number of timesteps for integration of $v_t$ | 3 |
| epL | Gradient descent step size: linear part of A | 2.00E-08 |
| epT | Gradient descent step size: translation part of A | 2.00E-01 |
| epv | Gradient descent step size: $v_t$ | 2.00E + 03 |
| pointsI | Landmark points for source image | None |
| pointsJ | Landmark points for target image | None |
| muB | Mean intensity/color of background pixels | None |
| muA | Mean intensity/color of artifact pixels | None |
| L | Initial guess for linear transform | None |
| T | Initial guess for translation | None |
| A | Initial guess for affine matrix. Either L and T can be specified, or A, but not both | None |

**Runtime estimate.** Runtime for the Stalign.LDDMM function was estimated for CPU settings using a MacBook Pro with an 2.4 GHz 8-Core Intel Core i9 processor and 32 GB 2400 MHz DDR4 memory, and for GPU settings using an Intel Xeon W-3365 2.7 GHz Thirty-Two Core 48MB 270 W processor with 8 x DDR4-3200 16GB ECC Reg memory and a NVIDIA RTX A5000 24GB PCI-E video card.

### Reporting summary

Further information on research design is available in the Nature Portfolio Reporting Summary linked to this article.

## Data availability

All data that was aligned with STalign is publicly available. MERFISH datasets are available on the Vizgen website for *MERFISH Mouse Brain Receptor Map data release* (https://info.vizgen.com/mouse-brain-map). The Visium dataset is available on the 10X Datasets website for *Spatial Gene Expression Dataset by Space Ranger 1.3.0* (https://www.10xgenomics.com/resources/datasets/adult-mouse-brain-ffpe-1-standard-1-3-0). STARMAP Plus data is available on the Broad Single Cell Portal (https://singlecell.broadinstitute.org/single_cell/study/SCP1830/spatial-atlas-of-molecular-[...]pes-and-aav-accessibility-across-the-whole-mouse-brain). Developing heart data is available on the Human Developmental Cell Atlas https://hdca-sweden.scilifelab.se/a-study-on-human-heart-development/ via ST_heart_all_detected_nuclei.RData from https://github.com/MickanAsp/Developmental_heart. The Xenium dataset (In Situ Replicate 1) of a fresh frozen mouse brain coronal section is available on the 10X Datasets website for *Mouse Brain Dataset Explorer* (https://www.10xgenomics.com/products/xenium-in-situ/mouse-brain-dataset-explorer). The two Xenium datasets (In Situ Replicate 1 and In Situ Replicate 2) of a single breast cancer FFPE tissue block are available on the 10X Datasets website for *High resolution mapping of the breast cancer tumor microenvironment using integrated single cell, spatial and* in

situ *analysis of FFPE tissue* (https://www.10xgenomics.com/products/xenium-in-situ/preview-dataset-human-breast). The CCF and brain region annotations are available from the Allen Brain Atlas API https://help.brain-map.org/display/mouseconnectivity/API. Source data are provided with this paper. Aligned cell positions, gene expression counts, cell-type annotations, and brain structure annotations generated in this study derived from these public datasets have been deposited in a Zenodo repository (https://doi.org/10.5281/zenodo.8384019)[34].

## Code availability

STalign is available as an open-source Python toolkit at https://github.com/JEFworks-Lab/STalign[35] with additional documentation and tutorials available at https://jef.works/STalign. Code (STalign.py and point_annotator.py) is also archived as Supplementary Software.

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

## Acknowledgements

K.C., M.A., G.A., L.A., and J.F. are supported by the National Institute of General Medical Sciences of the National Institutes of Health under Award Number R35-GM142889 and the National Science Foundation under Grant No. 2047611. M.A. and J.M.K. are supported by the National Institute of Drug Abuse under Award Numbers DP1-DA056668. Figure 5 and Supplementary Fig. 11 were created with component graphics licensed from Biorender.com.

## Author contributions

D.T. led the development of the STalign software and mathematical modeling with input from J.F., K.C., M.A., and M.I.M. J.F., K.C., and M.A. led the application of STalign to various ST datasets with input from D.T. K.C. evaluated the performance of STalign for 2D alignment under the guidance of J.F. M.A. evaluated the performance of STalign for 3D

alignment under the guidance of J.F. and J.M.K. O.K.A. evaluated the performance of STalign using landmark-based approaches under the guidance of D.T. G.A. performed runtime benchmarks and code revisions under the guidance of K.C., D.T., and J.F. L.A. contributed to the revision under the guidance of K.C. and J.F. All authors contributed to the writing of the manuscript. All authors approved the final manuscript.

## Competing interests

MIM is a founder of AnatomyWorks. This arrangement has been reviewed and approved by the Johns Hopkins University in accordance with its conflict-of-interest policies. The other authors declare no competing interests.
