## [Peer Review File · Nature Communications]

STalign: Alignment of spatial transcriptomics data using diffeomorphic metric mappingReviewer #1 (Remarks to the Author):

This manuscript presents a computational method, STalign, to align pairs of spatial omics data. A key novelty of STalign is that it formulates the problem of aligning spatial omics data in the large deformation diffeomorphic metric mapping (LDDMM) framework. LDDMM was mainly used in mapping dense medical images such as CT and MRI. As spatial omics data have increasing resolution and number of spatial locations, it is appropriate to adopt LDDMM framework in spatial omic alignment problem.

STalign assumes the input spatial omics have single-cell resolution and is applicable to MERFISH, Xenium, and H&E image data. It operates on cell locations (marginal space measure) rather than measured omics space (e.g. gene expression) and thus features cross-modality alignment. In addition, STalign also features partial alignment, handling nonlinear distortions, and 2D-3D alignment due to LDDMM framework. The 2D-3D alignment is also novel for spatial omics alignment. The results show decent alignment accuracy on alignments between pairwise MERFISH samples, MERFISH-H&E image alignments, and MERFISH to 3D common coordinate framework (CCF) alignments.

Despite the novelty of STalign, I have the following concerns and suggestions.

1. The manuscript lacks discussions and comparisons with previous ST alignment methods. Particularly, I don't see any alignment results from previous methods in the results, nor are they compared with STalign. Here are some popular methods I recommend to compare. Jones, Andrew, et al. "Alignment of spatial genomics and histology data using deep Gaussian processes." *BioRxiv* (2022): 2022-01. Zeira, Ron, et al. "Alignment and integration of spatial transcriptomics data." *Nature Methods* 19.5 (2022): 567-575. In addition, it's worth mentioning and discussing the work by Biancalani, Tommaso, et al. "Deep learning and alignment of spatially resolved single-cell transcriptomes with Tangram." *Nature methods* 18.11 (2021): 1352-1362., which has a section to find the closest image in CCF to a given H&E image.
2. Method related clarifications:
 - What does I model exactly in different technologies? On one hand, I is the rasterization function of marginal space measure (or cell densities) according to online method section 1.2. On the other hand, the target space I^T may be RGB image values instead of cell densities. Can I^S and I^T model different quantities (one cell densities and the other RGB) in the same optimization objective? I suggest clarify that in the method section.
 - If I function can map the spatial spots to RGB channels in H&E images, we can change I to map spatial spots to dimensionality-reduced expressions (e.g. PCs or glmPCs) for SRT data. Can you discuss whether STalign can be used to Visium data if using I to map to PCs of gene expression?
 - f_{θ} is a polynomial transformation of an image. Is it only applied to H&E images or also cell densities? What order of polynomial is used? Is there evidence from the data to justify the choice? And whether it may overfit the data so that the velocity v underfits?
 - There is a typo in M_{θ} term in Fig 1 and online method equation 14, the square bracket doesn't close. In the regularization term of velocity v , how to choose spatial smoothness coefficient a ?
3. Manual affine transformation is used in result section for comparison. However, it's unclear what criteria is used in "manual". In the most challenging evaluation setting, one could find an affine transformation by Procrustes analysis using landmarks, i.e., finding an affine transformation to best transform the coordinates of landmarks in one dataset to coordinates of the other. I wonder how STalign compares with the supervised affine transformation. If the samples are rigid enough, it's helpful for evaluating STalign as an unsupervised method; otherwise, it will show the superiority of modeling non-linearity in alignment.
4. Following the above point, the weight of regularization term of v may need to be adjusted when different levels of non-linear distortions are required for alignment. Is that a true claim? Is there any guidance for adjusting the regularization to account for the levels of non-linear distortions? Or is the regularization robust enough if there is evidence from simulations?
5. When evaluating the alignment using gene expressions, cosine similarity of individual genes is evaluated. Spatially varying genes tend to have high cosine similarity, and non-spatially varying

genes have low similarity. I wonder what the trend look like for the overall cellular expression states (PCs or glmPCs)?

6. Fig 5b is supposed to show that when aligning a 2D sample to 3D space, the 2D sample may not fall into a plane but a curve manifold. However, it is hard to see in the current plot. Is it possible to highlight the tissue region with the curve and to shift the angle of viewpoint to better show that?

7. Comment on code and documentation

`[-]` - The dependencies are not listed in the documentation (e.g. `nrnd`)`[-]`

- `pip install` command leads to incomplete installation for me. After `import STalign`, I only got "`__*__`" attributes and functions of `STalign` class.`[-]`

- `LDDMM` function allows to specify the device (`cpu` or `cuda`) in `PyTorch`, but the following functions (`build_transform` and `transform_image_atlas_to_target`) only allows `cpu`. It would be more convenient to automatically choose the device based on `LDDMM` function.`[-]`

- `analyze3Dalign` in `STalign.py` function has inconsistent variable name in the function argument vs function body.

- The Xenium data I downloaded ("Supplemental: Post-Xenium H&E image") has 100 times more pixels than the tutorial. It would be better to either point more specifically which file from 10X website contains the image that is used.

Reviewer #2 (Remarks to the Author):

The authors presented `STalign`, which can align samples of spatially resolved single cell gene expression data. This is one of the first methods for this problem, and is a very timely tool. The results are promising. However, there are several aspects the manuscript should be improved on:

1. The paper mentioned that differences between different samples can come from both biological variations and technical variations, as well as distortions between tissues. As these have not been discussed much before, it would be helpful if the authors can expand the discussion on this, which can also make the goal of `STalign` more clear. For example, where can technical variations come from? Is `STalign` designed to remove the technical variations and retain biological variations, and why?

2. The problem of aligning spatial omics samples is a relatively new problem and existing methods are rare, but this should be discussed in the introduction: what are existing practices if samples need to be aligned? The authors compared their results with affine transformation, and this should be included in the introduction as existing practices.

3. Fig. 5e and Supp Fig. 9a presented "brain regions" annotated after applying `STalign`. It is not clear what are the biological meanings of these brain regions and how these results help validate the results of `STalign`.

4. Fig. 5h presented entropy values of cell type compositions in two scenarios: original and expanded. It is not clear why one should expect the entropy to be low for the original scenario. Should we expect the distribution of cell types to be skewed in the original scenario, and why?

5. This is a general and crucial point for all the alignment results: so far the results focus on showing that they alignment is performed properly. However, one major goal of aligning samples is to find differences between samples. Such differences have not been discussed much in the results in Lines 142-146, much of it was attributed to noise. To show the application of `STalign` in helping to compare samples, it is suggested to give one or two examples of meaningful differences found between samples, such as genes that have different spatial patterns or different cell types distributions in space.

6. Fig. 6 is presented and discussed in the section Discussion. It should be re-organized to be included in the Results section.

7. In Fig. 2b: what are the genes for the three plots? One would guess they are the genes in Fig.

2a but there is no mention of this. Same for Fig. 3c.

Reviewer #3 (Remarks to the Author):

The alignment of spatial transcriptomics (ST) data across slices, samples, and technologies is an important topic in the field. The authors have proposed STalign, a tool that aligns ST datasets by considering tissue section variations and non-linear distortions using diffeomorphic metric mapping. The authors have demonstrated that STalign achieves improved gene expression and cell-type correspondence compared to manual and landmark-based alignments.

Major concerns:

1. The authors have demonstrated that STalign significantly reduces the root mean square error (RMSE) between landmarks in single-cell ST datasets within technologies compared to alternative methods. To provide a more comprehensive analysis, it would be beneficial for the authors to showcase the gene expression correspondence across matched spatial locations for all the compared methods, not just STalign.
2. The alignment performed by STalign relies on tissue structures formed by cell densities. In the case of imaging-based ST, cell segmentation is typically utilized to identify cells, introducing additional uncertainties. It would be important for the authors to discuss whether this step has any influence on the performance of STalign and address its potential impact.
3. As discussed by the authors, achieving a one-to-one correspondence between cells across samples is challenging, particularly for pathological tissues. In such scenarios, if landmark information is available, it would be interesting to explore whether STalign can incorporate known tissue structures or features to enhance alignment. Currently, the authors only explore these features to generate initial values, but further exploration or discussion could be beneficial.
4. After aligning spatial transcriptomics (ST) data across different platforms, we obtain gene expressions for matched spatial locations. However, it is important to consider that different ST platforms may have varying gene coverage, and batch effects can also be present. In light of these considerations, what types of downstream analyses could be conducted based on these matched spatial observations?
5. The authors have made a Python package with nice tutorials for various usage examples. However, it would be helpful to include a tutorial specifically addressing single-cell and spot-resolution ST alignment. Additionally, providing instructions on preparing a single-cell resolution HE staining image obtained from Visium would greatly assist researchers using this technique.

Minor concerns:

1. Line 183: The authors mention a "matching probability > 0.85 ." It would be helpful if the authors could provide clarification on how this quantity is calculated or derived.
2. It would also be beneficial if the authors could include information regarding the computational times taken for STalign, as this would provide insights into the tool's efficiency and practicality.

Point by Point Response – Overview

We sincerely thank the editor and the reviewers for their insightful and constructive feedback in helping us improve this manuscript. We have now revised the manuscript to address all the points raised by the reviewers, organized herein as a point-by-point response. Throughout this point-by-point response, reviewer comments are shown in **blue**, with our responses in **green**, and changes to the manuscript in **black**.

Reviewer #1 (Remarks to the Author):

This manuscript presents a computational method, STalign, to align pairs of spatial omics data. A key novelty of STalign is that it formulates the problem of aligning spatial omics data in the large deformation diffeomorphic metric mapping (LDDMM) framework. LDDMM was mainly used in mapping dense medical images such as CT and MRI. As spatial omics data have increasing resolution and number of spatial locations, it is appropriate to adopt LDDMM framework in spatial omic alignment problem.

STalign assumes the input spatial omics have single-cell resolution and is applicable to MERFISH, Xenium, and H&E image data. It operates on cell locations (marginal space measure) rather than measured omics space (e.g. gene expression) and thus features cross-modality alignment. In addition, STalign also features partial alignment, handling nonlinear distortions, and 2D-3D alignment due to LDDMM framework. The 2D-3D alignment is also novel for spatial omics alignment. The results show decent alignment accuracy on alignments between pairwise MERFISH samples, MERFISH-H&E image alignments, and MERFISH to 3D common coordinate framework (CCF) alignments.

Despite the novelty of STalign, I have the following concerns and suggestions.

1. The manuscript lacks discussions and comparisons with previous ST alignment methods. Particularly, I don't see any alignment results from previous methods in the results, nor are they compared with STalign. Here are some popular methods I recommend to compare. - Jones, Andrew, et al. "Alignment of spatial genomics and histology data using deep Gaussian processes." *BioRxiv* (2022): 2022-01. - Zeira, Ron, et al. "Alignment and integration of spatial transcriptomics data." *Nature Methods* 19.5 (2022): 567-575. In addition, it's worth mentioning and discussing the work by Biancalani, Tommaso, et al. "Deep learning and alignment of spatially resolved single-cell transcriptomes with Tangram." *Nature methods* 18.11 (2021): 1352-1362., which has a section to find the closest image in CCF to a given H&E image.

We thank the reviewer for the recommendation to compare our method to previous ST alignment methods and for providing references for these potential methods for comparison. As recommended by the reviewer, we have tried to compare STalign to PASTE, the tool introduced in Zeira, Ron, et al. "Alignment and integration of spatial transcriptomics data." *Nature Methods* 19.5 (2022): 567-575.

We were able to successfully install PASTE and use their tutorial to reproduce their alignment of four breast cancer ST datasets where each dataset consists of transcriptomic measurements for roughly 200 spatially gridded spots. However, we encountered difficulties when applying PASTE to larger datasets. We attempted to use PASTE to align the same MERFISH single cell resolution datasets analyzed previously in our Figure 1 with nearly 100,000 cells. However, we encountered errors because the process was too computationally intensive. In repeated attempts, the Python kernel died due to insufficient memory.

In their original manuscript, the authors of PASTE note that the algorithm takes $O(n^2n' + nn'^2)$ operations per iteration where n is the number of spatially resolved measurements (spots or cells) in the source and n' is the number of spatially resolved measurements (spots or cells) in the target. For our MERFISH data $n = 85958$ and $n' = 84172$ in contrast to the original datasets where n and n' were on the order of 100s. When we downsampled our data to only 20% of the cells ($n = 17900$ and $n' = 18113$), we were able to run PASTE with default parameters and no longer experienced the same insufficient memory errors. However, the alignment we achieved still visually appeared suboptimal (Reviewer Fig 1). PASTE returned a warning that 'numItermax reached before optimality. Try to increase numItermax', suggesting that the default parameters may not be optimal. However, increasing 'numItermax' did not lead to changes. As such, despite our best efforts, we were not able to achieve a reasonable alignment of MERFISH data using PASTE.

Reviewer Figure 1. Alignment with PASTE. A) Downsampled MERFISH data prior to PASTE alignment. B) After PASTE alignment with default parameters.

As described in Ron, et al., the authors of PASTE built their alignment tool to be applicable to “spatial transcriptomics” as defined by Stahl et al in 2016 as measuring RNA expression via a grid of 100s of barcoded spots. Spatial transcriptomics technologies have since evolved, particularly with imaging-based technologies such as MERFISH able to generate datasets with orders of magnitude more cells (approximately 10^5 to 10^6). In contrast, STalign is amenable to such spatial transcriptomics datasets with more cells.

For GPSA as described in Jones, Andrew, et al. "Alignment of spatial genomics and histology data using deep Gaussian processes", the authors similarly note that “While we find that GPSA finds more accurate alignments than [PASTE], this comes at the cost of time. Our variational inducing point inference reduces the time complexity from $O(n^3)$ to $O(nm^2)$, where n and m are the number of readouts and inducing points, respectively.” As such, the largest dataset analyzed in the paper was a Visium dataset with only 3,355 spots. As with PASTE, GPSA focuses on aligning data from pixel-resolution ST data within ST technologies.

While a direct comparison with PASTE and GPSA could not be established, we have included a mention of this difference between STalign and these methods in both the introduction and discussion, with relevant excerpts provided below for the reviewer’s reference:

INTRODUCTION

(line 56) Considering the recent development of such ST technologies, options for spatially aligning ST datasets are still limited. Previous computational methods have focused on spatial alignment of ST datasets for which each dataset is assayed using the same pixel-resolution ST technology with only a few hundred to a few thousand spatial measurements^{1,2}. These methods face challenges in scaling to larger, single-cell resolution ST datasets with tens to hundreds of thousands of spatial measurements. Further, spatial alignment of datasets across different ST technologies remains challenging. Other alignment methods are limited to rigid, affine transformation such as based on landmarks³ and cannot accommodate non-linear distortions.

...

DISCUSSION

(line 541) As ST technologies continue to evolve, we anticipate STalign will continue to be applicable due to our use of rasterization to convert the positions of single cells into an image with specified size and pixel resolution. The performance of each iteration of the STalign alignment algorithm scales with respect to the number of pixels in the rasterized image. Whereas other alignment algorithms generally scale in memory and runtime with the number of spatially resolved measurements (spots or cells)^{1,2}, which will likely make them computationally untenable as ST technologies evolve to increase the number of spatially resolved measurements that can be assayed. Overall, we anticipate that the ability for users to choose the pixel resolution, and therefore the number of pixels in the rasterized image, will allow STalign to maintain its utility for larger datasets.

With respect to Biancalani, Tommaso, et al.⁴, as the reviewer notes, the authors use Siamese Neural Networks to identify which coronal brain slice in a CCF best matches a given histology image. However, they do not use this match to create an alignment but rather to match tissue sources for single-cell RNA-seq datasets to multi-cellular spot-resolution spatial datasets. The alignment achieved by Tangram is then between single cells and spatial spots based on similarity in gene expression. In general, we find that it is worth mentioning how STalign relates to such gene-expression-based alignment tools and have therefore added this point to the discussion, provided below for the reviewer's reference.

(line 534) Likewise, STalign complements gene-expression-based approaches for sample alignment by focusing on the real space rather than a higher-order transcriptomic manifold.

2. Method related clarifications:

- What does \mathcal{I} model exactly in different technologies? On one hand, \mathcal{I} is the rasterization function of marginal space measure (or cell densities) according to online method section 1.2. On the other hand, the target space \mathcal{I}^T may be RGB image values instead of cell densities. Can \mathcal{I}^S and \mathcal{I}^T model different quantifies (one cell densities and the other RGB) in the same optimization objective? I suggest clarify that in the method section.

We thank the reviewer for the opportunity to clarify this important point. The reviewer is correct that \mathcal{I}^S and \mathcal{I}^T may model different quantifies.

We currently state in the Online Methods (section 1.5 Image Registration): Note that \mathcal{I}^T need not correspond to a smooth density image as defined in section 1.2. For example, we include the case where it is a red green blue image corresponding to an H&E stain. The function f_θ is a transformation of image contrast with unknown parameters θ . We use a polynomial for f_θ , in which case the minimizing parameters θ can be found exactly by solving a weighted least squares problem. The purpose of this transformation is to model differences in contrast between images from the same modality due to calibration issues; and contrast/color differences between different modalities.

As the reviewer suggested, we have now added further clarification earlier in the Online Methods (section 1.2. Rasterization) of the methods section with the added text provided below for the reviewer's reference: In this section we showed how a rasterized image \mathcal{I} can be produced from a list of cell location, in a manner compatible with the theory of varifolds. However, our registration algorithm can be performed with any standard rasterized image type. For example, in the main manuscript we show examples where \mathcal{I}^T is a red-green-blue image corresponding to a brightfield microscopy image of H&E stained tissue. How such images of different contrast profiles are handled is described in section 1.5.

- If \mathcal{I} function can map the spatial spots to RGB channels in H&E images, we can change \mathcal{I} to map spatial spots to dimensionality-reduced expressions (e.g. PCs or glmPCs) for SRT data.

Can you discuss whether STalign can be used to Visium data if using \mathcal{I} to map to PCs of gene expression?

The reviewer is correct in that we can change \mathcal{I} to incorporate gene expression information such as via dimensionality-reduced representations such as via PCs of gene expression.

In this work, we used only information about cell density in ST data to perform image registration, and reserved gene expression information for validation of the registration results. We anticipate the incorporation of gene expression information may improve registration accuracy for tissue regions with homogenous cell density but heterogeneous gene expression and cell-type composition.

However, we note that incorporating gene expression in the synthesis of \mathcal{I} will necessitate orthogonal means of validating registration results. As such, the incorporation of gene expression in the synthesis of \mathcal{I} , while beyond the scope of this paper, will be indeed an important area of method development in the future. We have now included this important point to the revised discussion, provided below for the reviewer's reference:

(line 595) Likewise, although we focused here on aligning based on cell densities, STalign and the underlying LDDMM framework can also be applied to align using cellular features such as gene expression magnitude, reduced dimensional representations of gene expression such as via principal components, or cell-type annotations, which may improve the accuracy of alignment for regions with homogenous cell density but heterogeneous gene expression and cell-type composition.

- f_{θ} is a polynomial transformation of an image. Is it only applied to H&E images or also cell densities? What order of polynomial is used? Is there evidence from the data to justify the choice? And whether it may overfit the data so that the velocity v underfit?

The reviewer is correct that f_{θ} is a polynomial transformation of an image. To clarify, we only apply the polynomial to the cell density image to estimate the contrast of the H&E image. We have now included the following clarification in the Online Methods (section 1.5):

In this work we found that first order polynomials were sufficient for accurate image registration. In other work in neuroimaging, we have used 3rd order polynomials, which have enough degrees of freedom to map the intensity of gray matter, white matter, and background to arbitrary intensities. Because there are many more pixels than degrees of freedom, it is unlikely that these polynomials will overfit the observed data \mathcal{I}^T . However, depending on the initialization of transformation parameters this is possible: if tissue in \mathcal{I} and \mathcal{I}^T do not overlap at all, parameters θ may be estimated to zero out imaging information and transform \mathcal{I} into a constant function that looks like background only.

- There is a typo in $\$M_{\{\theta\}}\$$ term in Fig 1 and online method equation 14, the square bracket doesn't close.

We thank the reviewer for bringing this error to our attention. We have corrected the equation in the online methods and in Fig 1, provided below for the reviewer's reference:

- In the regularization term of velocity $\$v\$,$ how to choose spatial smoothness coefficient $\$a\$$?

In STalign, the spatial smoothness coefficient $\$a\$$ is a user defined parameter with units of length. The user can choose the scale of $\$a\$$ based on the resolution of the images to be aligned. We note that small values of $\$a\$$ may be overfitting of noise whereas large values of $\$a\$$ may lead to low accuracy. We have now included a mention of this in the Online Methods (section 1.5): Note that small values of $\$a\$$ may be overfitting of noise whereas large values of $\$a\$$ may lead to low accuracy.

While users could estimate $\$a\$$ from populations as in Zhang, Miaomiao et al. "Bayesian estimation of regularization and atlas building in diffeomorphic image registration." *Information processing in medical imaging : proceedings of the ... conference* vol. 23 (2013): 37-48 doi:10.1007/978-3-642-38868-2_4 and Tward, Daniel J et al. "Robust Diffeomorphic Mapping via Geodesically Controlled Active Shapes." *International journal of biomedical imaging* vol. 2013 (2013): 205494. Doi:10.1155/2013/205494, we believe this to be likely unnecessary for general practical usage.

Therefore, to help users choose an appropriate λ , we have now provided Jupyter notebooks demonstrating the impact of different λ for diverse datasets to assist with this choice. For example, in <https://jef.works/STalign/notebooks/heart-alignment.html>, we use $\lambda = 250$ instead of the default value, $\lambda = 500$ to improve alignment accuracy.

3. Manual affine transformation is used in result section for comparison. However, it's unclear what criteria is used in "manual". In the most challenging evaluation setting, one could find an affine transformation by Procrustes analysis using landmarks, i.e., finding an affine transformation to best transform the coordinates of landmarks in one dataset to coordinates of the other. I wonder how STalign compares with the supervised affine transformation. If the samples are rigid enough, it's helpful for evaluating STalign as an unsupervised method; otherwise, it will show the superiority of modeling non-linearity in alignment.

We thank the reviewer for the suggestion of comparing STalign with a supervised affine transformation that optimizes correspondence of landmark coordinates. We have now added this comparison. Specifically, we derive such a supervised affine transformation by registering manually placed landmarks in the source to matched manually placed landmarks in the target shown in Supplemental Figure 1 for the Slice 2 Replicate 3 to Slice 2 Replicate 2 MERFISH datasets. We still find that STalign performs better than this supervised affine transformation with regards to spatial gene expression correspondence, suggesting that using STalign to solve a non-linear transformation results in superior alignment accuracy (Supplemental Figure 4). We hope that these revised results help clarify the superiority of modeling non-linearity in alignment as suggested by the reviewer and have updated the Results, the Methods, and Supplemental Figure 4 with revised excerpts included below:

RESULTS (line 179)

We next compare the alignment achieved with STalign to the alignment from a **supervised affine** transformation based on our previously manually placed landmarks (**Supp Fig 4a**, Methods). We visually confirm that a supervised affine alignment results in a lower degree of spatial gene expression correspondence than alignment by STalign (Supp Fig4b). We again evaluate performance of the supervised affine transformation using a pixel-based cosine similarity quantification (Supp Fig4c). We find that for spatially patterned genes, the cosine similarity is consistently higher with a mean difference of **0.09** for the alignment by STalign compared to supervised affine (**Supp Fig 4d**). In contrast, for non-spatially patterned genes, the cosine similarity is more comparable with a mean difference of **0.02** for the alignment by STalign compared to supervised affine (**Supp Fig 4e**). This greater improvement in spatial gene expression correspondence for the alignment achieved with STalign compared to supervised affine transformation for spatially patterned genes suggests that modeling non-linearity in alignment with approaches like STalign can achieve a higher alignment accuracy compared to linear alignment approaches.

METHODS (line 1006)

Comparison to supervised affine alignment of single-cell resolution ST datasets within technologies

In addition to alignment by STalign, we performed supervised affine alignment of Slice 2 Replicate 3 and Slice 2 Replicate 2 from the MERFISH datasets, with the former as the source and the latter as the target. **We manually placed 13 landmarks in the source and target that could be reproducibly identified (Supp Fig 1, Supp Table 1) using our script point_annotator.py. We solved for the affine transformation that minimized the error between these landmarks using least squares and applied the affine transformation to the cell positions of the source.** With the supervised affine aligned source and target, we repeated the expression-based performance evaluation described in section “Expression based performance evaluation for STalign-based alignment of single-cell resolution ST datasets within technologies.”

Supplemental Figure 4

Supplemental Figure 4. Evaluation of STalign against **supervised affine** alignment. **a.** Spatial agreement of target and source that has been aligned based on a simple affine transformation based on manually placed landmarks. **b.** Correspondence of gene expression spatial organization between the target and supervised affine aligned source for select spatially patterned genes. **c.** Transcript counts in the target compared to the supervised affine aligned source at matched pixels for select genes: *Cckar*, *Htr5b*, *Gabbr2* and *Ackr2*. **d.** Cosine similarities between

transcript counts in target versus aligned source for STalign compared to affine alignment for 457 spatially patterned genes. (mean difference = **0.09**) Genes featured in Supplemental Figure 4b-c, Figure 2a-c, and Supplemental Figure 2 are highlighted. e. Cosine similarities between transcript counts in target versus aligned source for STalign compared to manual alignment for 192 non-spatially patterned genes. (mean difference = **0.02**) Genes featured in Figure 2d-f and Supplemental Figure 3 are highlighted.

4. Following the above point, the weight of regularization term of v may need to be adjusted when different levels of non-linear distortions are required for alignment. Is that a true claim? Is there any guidance for adjusting the regularization to account for the levels of non-linear distortions? Or is the regularization robust enough if there is evidence from simulations?

The reviewer is correct that the weight of regularization term may need to be adjusted when different levels of non-linear distortions are required for alignment. In our computational implementation, the user can tune the weight by utilizing σ^2_R . We have updated our regularization equation in the Online Methods (section 1.5 Image registration) to reflect the presence of this parameter.

$$R(v) = \frac{1}{2\sigma_R^2} \int_0^1 \int_{\mathbb{R}^D} |(\text{id} - a^2 \Delta)^p v_t(x)|^2 dx dt$$

As with the smoothness coefficient α , we have the same sort of issue where large values may lead to overfitting, and small values result in a smoother transformation but may lead to inaccuracy. We have now included a mention of this in the Online Methods (section 1.5 Image registration) with relevant excerpts provided below for the reviewer's reference:

σ^2_R is a user tunable parameter that adjusts balance between matching accuracy and regularization, where large values correspond to less regularization and higher matching accuracy, and small values correspond to more regularization and lower matching accuracy.

Again, appropriate values can be estimated by fine tuning and observing the results or by comparing to the Jupyter notebook examples that we now provide.

5. When evaluating the alignment using gene expressions, cosine similarity of individual genes is evaluated. Spatially varying genes tend to have high cosine similarity, and non-spatially varying genes have low similarity. I wonder what the trend look like for the overall cellular expression states (PCs or glmPCs)?

We thank the reviewer for the interesting suggestion. To explore this similarity in overall cellular expression states between aligned MERFISH and Visium samples, we performed the following analysis:

As done previously, to enable comparison between the single cell resolution MERFISH data with the multi-cellular spot resolution Visium data, we collapsed the MERFISH data into pseudospots, where each pseudospot has a corresponding Visium spot at the same position after alignment of the two samples using STalign. We subsequently performed PCA on the MERFISH pseudospots and obtained PC scores for the MERFISH pseudospots. We then applied the gene loadings to the Visium spots to obtain PC scores for the Visium spots in the same PC space as the MERFISH pseudospots.

To compare similarity of overall cellular expression states between MERFISH pseudospots and Visium spots at the same position, we evaluated the correlation of the PC scores between MERFISH pseudospots and Visium spots at the same position for the top PCs. Here, we hypothesize that good alignment would result in similar cellular expression states between MERFISH pseudospots and Visium spots at the same position. This would in turn be reflected in similar PC scores between MERFISH pseudospots and Visium spots at the same position in the joint PC space.

Looking in PC space (Review Figure 2A), we find that in the first PC, spots and pseudospots are separated by technology, with a subset of MERFISH pseudospots having lower scores on PC1 than other MERFISH pseudospots and most Visium spots. Such separation is not apparent in subsequent PCs.

We find that the correlation between the PC scores for PC1 and PC2 is relatively low compared to PC3 and PC4, which have relatively higher correlations (Reviewer Figure 2B).

Looking at spot scores in spatial coordinates (Reviewer Figure 2C), we see that the PCs are capturing aspects of the tissue architecture. For example, looking at the scores of the MERFISH pseudospots and Visium spots in PC1, we can see that spots in the dentate gyrus region have relatively higher scores than surrounding spots. However, these trends do not always correspond when looking at the same PC across the two technologies. For example, PC2 scores for Visium spots in the cortex region are relatively lower than other spots in the Visium sample, whereas corresponding pseudospots in the MERFISH sample have more intermediate scores on PC2.

These results indicate that technology-driven sources of variation in the data may be confounding this analysis. This could be reflective of gene-specific differences in RNA detection/capture efficiency between the technologies or differences in permeabilization efficiency across cell types or tissue regions between the two technologies. Ultimately, these effects render this analysis difficult to interpret. Since the same genes may not contribute to the same PCs across both technologies to the same extent, using PCs to evaluate similarity in gene expression states may not be reflective of good alignment but rather of technology-specific differences.

We believe our current alignment using cell-type proportion similarity via clustering and deconvolution analysis (Fig 4) provides a more interpretable evaluation of the overall cellular expression state.

Reviewer Figure 2. Principal component score correspondence between Visium spots and MERFISH pseudospots. A) MERFISH pseudospots and projected Visium spots in PC space in the first four PCs. B) Correspondence of PC scores for the first four PCs between MERFISH pseudospots and Visium spots in the same spatial position after alignment by STalign. C) PC scores of MERFISH pseudospots (top) and Visium spots (bottom) in spatial coordinates.

6. Fig 5b is supposed to show that when aligning a 2D sample to 3D space, the 2D sample may not fall into a plane but a curve manifold. However, it is hard to see in the current plot. Is it possible to highlight the tissue region with the curve and to shift the angle of viewpoint to better show that?

We thank the reviewer for the suggestion showing how the 2D slice was placed with respect to the 3D atlas. We have now updated Fig5b to highlight how the 2D sample does not fall into a plane but a curve manifold using an angled viewpoint as the reviewer suggested:

7. Comment on code and documentation

- The dependencies are not listed in the documentation (e.g. nrrd)

We thank the reviewer for taking the time to review and test out our code. We have now included a **requirements.txt** file in the STalign Github repo that includes all the package dependencies.

- pip install command leads to incomplete installation for me. After import STalign, I only got “__*__” attributes and functions of STalign class.

We have addressed the incomplete installation issue by adding a new **requirements.txt** and have updated the import instructions in the **README.md**. We have provided a screenshot of the updated import instructions below for the reviewer’s reference:

Installation & Import

Installation using pip

This installation method is intended for users who sets up a Python environment without `pipenv`.

```
$ pip install "git+https://github.com/JEFworks-Lab/STalign.git"
```

All dependencies will be installed into your selected environment with the above command. Dependencies can be found in the `requirements.txt` file.

Installation using Pipfile from source

This installation method is intended for users who sets up a Python environment with `pipenv`. `pipenv` allows users to create and activate a virtual environment with all dependencies within the Python project. For more information and installation instructions for `pipenv`, see <https://pipenv.pypa.io/en/latest/>.

Fork and `git clone` the `STalign` github repository.

From the base directory of your local `STalign` git repo, create a `Pipfile.lock` file from `Pipfile` using:

```
$ pipenv lock
```

NOTE: Since `Pipfile.lock` is platform-dependent and different across operating systems, do not commit `Pipfile.lock` to the git repo if contributing to `STalign` or collaborating with other people.

Install `PyTorch` dependency using:

```
$ pipenv install torch==2.0.0
```

Activate the virtual environment using:

```
$ pipenv shell
```

Deactivate the virtual environment using:

```
$ exit
```

Import

To import `STalign` into your Python script, use:

```
from STalign import STalign
```

- LDDMM function allows to specify the device (cpu or cuda) in PyTorch, but the following functions (build_transform and transform_image_atlas_to_target) only allows cpu. It would be more convenient to automatically choose the device based on LDDMM function.

We thank the reviewer for this catch and suggestion. We agree it would be more convenient for the user to more automatically specify the device used. We have therefore added various code blocks to our Jupyter notebook tutorials to assist in this automation. Some relevant code snippets are included below for the reviewer's convenience.

```
# set device for building tensors
if torch.cuda.is_available():
    torch.set_default_device('cuda:0')
else:
    torch.set_default_device('cpu')

# compute initial affine transformation from points
AI= STalign.transform_image_atlas_with_A(A, [YI,XI], I, [YJ,XJ])

fig,ax = plt.subplots(1,2)

if AI.is_cuda:
    ax[0].imshow((AI.cpu().permute(1,2,0).squeeze()), extent=15xtent)
else:
    ax[0].imshow((AI.permute(1,2,0).squeeze()), extent=15xtent)
ax[1].imshow((J.transpose(1,2,0).squeeze()), extent=15xtent)

ax[0].set_title('source with affine transformation', fontsize=15)
ax[1].set_title('target', fontsize=15)

# apply transform
15xt = STalign.build_transform(xv,v,A,XJ=[YJ,XJ],direction='b')
15xt = STalign.transform_image_atlas_to_target(xv,v,A,[YI,XI],I,[YJ,XJ])
phiipointsI = STalign.transform_points_atlas_to_target(xv,v,A,pointsI)

# plot with grids
fig,ax = plt.subplots()
levels = np.arange(-100000,100000,1000)

if 15xt.is_cuda:
    ax.contour(XJ,YJ,15xt[... ,0].cpu(), colors='r',linestyles='- ',
    \,levels=levels)
    ax.contour(XJ,YJ,15xt[... ,1].cpu(), colors='g',linestyles='- ',
    \,levels=levels)
else:
    ax.contour(XJ,YJ,15xt[... ,0], colors='r',linestyles='- ',levels=levels)
    ax.contour(XJ,YJ,15xt[... ,1], colors='g',linestyles='- ',levels=levels)
ax.set_aspect('equal')
ax.set_title('source to target')
```

```

if 16xt.is_cuda:
    ax.imshow(16xt.cpu().permute(1,2,0)/torch.max(16xt.cpu()), extent=16xtent)
else:
    ax.imshow(16xt.permute(1,2,0)/torch.max(16xt), extent=16xtent)

if phiipointsI.is_cuda:

ax.scatter(phiipointsI[:,1].cpu().detach(), phiipointsI[:,0].cpu().detach(), c=
"m")
else:
    ax.scatter(phiipointsI[:,1].detach(), phiipointsI[:,0].detach(), c="m")

```

- analyze3Dalign in STalign.py function has inconsistent variable name in the function argument vs function body.

We thank the reviewer for taking the time to review and test out our code. We changed the variable names passed into the function to match the names in the body of the function.

Specifically, we changed:

```

def analyze3Dalign(labelfile, xv, v, A, XJ, dx, scale_x, scale_y, x, y, X_, Y_,
namesdict, device='cpu'):

```

to:

```

def analyze3Dalign(labelfile, xv, v, mat, xJ, dx, scale_x, scale_y, x, y, X_, Y_,
namesdict, device='cpu'):

```

- The Xenium data I downloaded (“Supplemental: Post-Xenium H&E image”) has 100 times more pixels than the tutorial. It would be better to either point more specifically which file from 10X website contains the image that is used.

The reviewer is correct that the online tutorial uses an H&E image that is smaller than the high resolution TIFF that is available through the 10X website. The image was resized for the online tutorial in order to minimize storage on Github. We have now clarified this in the online tutorial (<https://jef.works/STalign/notebooks/xenium-heimage-alignment.html>) and further specify the specific file from the 10X website that contains the image used as the reviewer suggested. The text is provided below for the reviewer’s reference.

To obtain the single cell spatial transcriptomics data, we can download the Xenium Output Bundle from the 10X website: <https://www.10xgenomics.com/products/xenium-in-situ/preview-dataset-human-breast> Expanding the downloaded Xenium_FFPE_Human_Breast_Cancer_Rep1_outs.zip, we will use the single cell positions stored in cells.csv.gz. Likewise, we can download the accompanying H&E staining image Supplemental: Post-Xenium H&E image (TIFF) as Xenium_FFPE_Human_Breast_Cancer_Rep1_he_image.tif.

To reproduce this tutorial, we have placed these files in a folder called `xenium_data/` with `cells.csv.gz` renamed as `Xenium_FFPE_Human_Breast_Cancer_Rep1_cells.csv.gz` for organizational purposes. Likewise, to minimize storage, for this tutorial, we have resized the high resolution H&E TIF image into a smaller PNG image as `Xenium_FFPE_Human_Breast_Cancer_Rep1_he_image.png`

Reviewer #2 (Remarks to the Author):

The authors presented STalign, which can align samples of spatially resolved single cell gene expression data. This is one of the first methods for this problem, and is a very timely tool. The results are promising. However, there are several aspects the manuscript should be improved on:

1. The paper mentioned that differences between different samples can come from both biological variations and technical variations, as well as distortions between tissues. As these have not been discussed much before, it would be helpful if the authors can expand the discussion on this, which can also make the goal of STalign more clear. For example, where can technical variations come from? Is STalign designed to remove the technical variations and retain biological variations, and why?

We thank the reviewer for the opportunity to clarify this important point. As an example, consider we are interested in characterizing the molecular differences between case and control brains at spatially matched locations. We can apply spatial transcriptomics to thin sections of our case and control brains to achieve such spatial molecular profiling. However, appropriately matching the spatial locations between case and control brains is integral to identifying condition-specific molecular differences. For example, if we observe molecular differences between the hippocampus brain region in the case brain and the cortex brain region in the control brain, these observed differences would likely be driven by region-specific differences rather than a difference between case and control.

As the reviewer notes, structural variations across tissue samples may come from both biological and technical sources. When obtaining these thin sections of brains, the experimental process may introduce technical variation. For example, these thin sections may be subject to different linear distortions, such as being placed at a rotated angle, as well as nonlinear distortions, such as compacting and stretching (Reviewer Fig 3A). Different brains may also exhibit structural variation for biological reasons due to natural inter-individual differences. Again, if such variation is not accounted for, making spatial molecular comparisons would be very difficult because perceived molecular differences may simply be due to poor alignment (Reviewer Fig 3B).

Reviewer Figure 3. A) Cartoons of control and case brain slices. Note the case brain slice was subject to different linear and non-linear distortions of tissue structures. B) Even with affine alignment of the case to control brain slices, observed molecular variation is confounded by lack of structural alignment.

Thus, in order to make spatial molecular comparisons at matched spatial locations, assuming that case and control brains are structurally similar, we must structurally align these 2 brain sections. We hope this clarifies where technical variations can come from and the goal of STalign. We have incorporated these points into the introduction with an excerpt provided below for the reviewer's reference:

(line 39) Spatial transcriptomics (ST) technologies have enabled high-throughput, quantitative profiling of gene expression within individual cells and small groups of cells in fixed, thin tissue sections. Comparative analysis of ST datasets at matched spatial locations across tissues, individuals, and samples provides the opportunity to interrogate spatial gene expression and cell-type compositional variation in the context of health and disease. Such comparative analysis is complicated by technical challenges such as in sample collection, where the experimental process may induce tissue rotations, tears, and other structural distortions. Other challenges include biological variation such as natural inter-individual tissue structural differences. In order to reliably characterize spatial molecular differences between ST datasets along comparative axes of interest, it is integral to control for potentially confounding tissue structural variation by spatially aligning these tissue structures across ST datasets.

2. The problem of aligning spatial omics samples is a relatively new problem and existing methods are rare, but this should be discussed in the introduction: what are existing practices if samples need to be aligned? The authors compared their results with affine transformation,

and this should be included in the introduction as existing practices.

We thank the reviewer for the suggestion. We agree that the problem of aligning spatial omics samples is a relatively new problem given the novelty of the technologies so existing methods are rare. We have now revised the introduction to include a mention of other existing alignment approaches including affine transformation as suggested by the reviewer, provided below for the reviewer's reference:

(line 56) Considering the recent development of such ST technologies, options for spatially aligning across ST datasets are still limited. Previous computational methods have focused on spatial alignment of ST datasets assayed using the same pixel-resolution ST technology with only a few hundred spatial measurements¹. As such, these methods are not scalable to larger, single-cell resolution ST data. Likewise, spatial alignment of datasets across samples and technologies remains challenging. Other alignment methods are limited to rigid, affine transformation such as based on landmarks² and cannot accommodate non-linear distortions.

3. Fig. 5e and Supp Fig. 9a presented “brain regions” annotated after applying STalign. It is not clear what are the biological meanings of these brain regions and how these results help validate the results of STalign.

We thank the reviewer for the opportunity to clarify this point. In the construction of the Allen Brain Atlas CCF, brain regions were delineated based on several features like cellular architecture, differential gene expression, and functional properties via modalities such as histological stains, in situ hybridization, and connectivity experiments to generate a set of reference brain region annotations. By aligning our spatial transcriptomic datasets to the Allen Brain Atlas, we were able to annotate cells in our spatial transcriptomics data with these reference brain region annotations based on proximity.

Numerous prior studies have shown that brain regions can be identified based on the upregulation of particular genes. We use this characteristic gene expression to confirm that the brain regions annotated using this alignment to the Allen Brain Atlas did indeed contain expression of known transcriptomic markers, supported by prior literature. For example, the gene *Grm2* is highly expressed in the dentate gyrus and has low or absent expression in other brain regions. We can therefore use these known region-specific gene expression patterns to confirm that the brain regions labeled by STalign indeed contain the expected genes, supported by prior literature. As shown in the paper, the high expression of *Grm2* almost completely overlaps with the region defined as the dentate gyrus (obtained from applying STalign), showing that the transcriptomic profile of the dentate gyrus is consistent with prior literature. This analysis is repeated in Fig. 5e and Supp Fig. 9a with other genes known to be expressed in brain regions, showing the applicability of this phenomenon across brain regions of different sizes and shapes and its consistency across brain slice replicates. If regions annotated using this alignment to the Allen Brain Atlas were improperly labeled, we would not expect the region-

associated genes to be expressed only in the region of interest. Visualizing expression of known gene-region pairs confirms that the alignment of the slice to the Allen Atlas is reasonable.

We have clarified these points in the revised manuscript, provided below for the reviewer's reference:

(line 303) In the construction of the Allen Brain Atlas CCF, brain regions were delineated based on several features like cellular architecture, differential gene expression, and functional properties via modalities such as histological stains, in situ hybridization, and connectivity experiments to generate a set of reference brain region annotations. By aligning to this CCF, we can lift over these annotations to each cell (Fig 5c, Supp Fig 8a), enabling further evaluation of variations of gene expression and cell-type composition within and across these annotated brain regions.

(line 341) Numerous previous studies have shown that some brain regions can be demarcated based on the expression of particular genes. We use these characteristic gene expression patterns to evaluate whether the brain regions lifted over by STalign indeed contain expression of known marker genes. Consistent with previous studies, we found *Grm2* to be visually primarily enriched in the dentate gyrus brain region¹¹, *Sstr2* to be enriched in cerebral cortical layers 5 and 6 brain region¹², and *Gpr161* to be enriched in the CA1 brain region¹³ (Fig 5f), which was consistent across replicates (Supp Fig 9b).

4. Fig. 5h presented entropy values of cell type compositions in two scenarios: original and expanded. It is not clear why one should expect the entropy to be low for the original scenario. Should we expect the distribution of cell types to be skewed in the original scenario, and why?

We thank the reviewer for taking the opportunity to clarify this point. Many brain regions have a characteristic cell type distributions. For example, in the striatum brain region, approximately 95% of neurons are medium spiny neurons, layer 1 of the neocortex only contains inhibitory neurons, and in the hippocampus, 90% of hippocampal neurons are excitatory pyramidal neurons. Due to these characteristic cell-type distributions in which one or a few cell-types predominate a brain region, we would expect the cell-type entropy for these brain regions to be low. Cell type entropy is high when all cell types are equally likely and low when only a few cell types populate the region, according to the formula $\sum p(x) * \log(p(x))$ where $p(x)$ is the probability of picking cell-type x from the given brain region. We use this principle of cell-type entropy to analyze whether boundaries of brain regions are accurately defined. Increasing the boundaries of the brain region should result in higher entropy, as the characteristic cell-type composition is lost, which we confirm with our analyses. If STalign-annotated regions are not well-defined, we would not expect to see a significant change in cell-type entropy in the original versus expanded brain regions due to the absence of the characteristic cell-type composition in both cases.

We have clarified these points in the revised manuscript, provided below for the reviewer's reference:

(line 348) Next, we took a more agnostic approach to assess the performance of our atlas alignment and lift-over annotations by evaluating the consistency of cell-type compositional heterogeneity within brain structures across replicates. To identify cell-types, we perform unified transcriptional clustering analysis on these 9 ST datasets to identify transcriptionally distinct cell clusters and annotate them as cell-types based on known differentially expressed marker genes (Methods, Fig 5e-f, Supp Fig 9a). Many brain regions are known to have a characteristic cell type distribution¹⁷⁻¹⁹. Consistent with previous studies²⁰, we observed cell-types to be spatially and compositionally variable across brain regions (Fig5c, Fig 5e).

...

(line 386) Finally, we also sought to assess the performance of our atlas alignment and lift-over annotations by evaluating cell-type compositions within and beyond annotated brain region boundaries (Methods). Specifically, we calculated the entropy of each brain region based on the region's cell-type composition compared to if the boundaries of these regions were expanded (Fig 5h). Again, due to the characteristic cell-type distributions within brain regions in which one or a few cell-types predominate, we would expect accurate lift-over brain region annotations to exhibit entropies that are comparatively lower than if the boundaries of these regions were expanded, as more cell-types would be incorporated into the region and entropy would increase.

5. This is a general and crucial point for all the alignment results: so far the results focus on showing that they alignment is performed properly. However, one major goal of aligning samples is to find differences between samples. Such differences have not been discussed much in the results in Lines 142-146, much of it was attributed to noise. To show the application of STalign in helping to compare samples, it is suggested to give one or two examples of meaningful differences found between samples, such as genes that have different spatial patterns or different cell types distributions in space.

We thank the reviewer for the opportunity to clarify how STalign may help in comparing meaningful differences between ST samples. Several studies have been published comparing different samples assayed with ST technologies to extract meaningful differences between samples.

As examples, a study by Allen et al (2023) used the spatial transcriptomics technology MERFISH to characterize age-related gene expression changes to investigate functional decline during aging. They probed for approximately 400 genes in 1, 6, and 21 month old mice, and compared spatial gene expression profiles across these time points, revealing novel findings regarding spatially-dependent changes in astrocytic and microglial states in young versus older mice in different brain regions. To perform this comparison, the authors performed brain region segmentation by clustering cells based on their transcriptomic profiles and local spatial

neighborhoods. STalign could provide an alternative method to defining these brain regions, by aligning brain slices to the Allen Reference Atlas, which eliminates the potential biases and circularity of defining these regions based on their transcriptomic profiles, and subsequently analyzing transcriptomic profiles within regions.

Another study by Ferriera et al (2021) used the spatial transcriptomics technology Visium to investigate the spatial gene expression programs involved in acute kidney injury (AKI) by comparing different durations of injury and recovery at early acute (4 hours, 12 hours), acute (2 days), and late (6 weeks) time points in murine models of AKI as well as a control murine kidney. This study identified differentially expressed genes through the course of injury and recovery on the whole tissue level. Alternatively, using STalign, these sections could be aligned to each other, and differentially expressed genes could have been identified for distinct spatial aligned spatial locations in the tissue.

We have now added to the discussion these two examples of meaningful differences found between samples that could be identified through spatial alignment, provided below for the reviewer's reference:

(Line 537) We further anticipate future applications of STalign to ST data from structurally matched tissues in case-control settings will enhance the throughput for yielding meaningful comparisons regarding gene expression and cell-type distributions in space as evidenced by recent applications of ST technologies to characterize spatially-resolved age-related²⁶ and injury-related²⁷ gene expression variation.

6. Fig. 6 is presented and discussed in the section Discussion. It should be re-organized to be included in the Results section.

We thank the reviewer for the suggestion to present Fig. 6 in the Results section. We have updated Fig.6 and moved it from the Discussion section into the Results section. The figure and excerpt of text are provided below.

(line 411) STalign applicable to diverse tissues profiled by diverse ST technologies

STalign relies on variation in cell densities that generally form visible structures that can be used for alignment. As we have shown, alignment across samples and animals is possible for tissues with highly prototypic structures such as the brain. We further highlight the applicability of STalign to the diverse ST technologies that can assay this tissue by demonstrating that we can apply STalign to achieve structural correspondence for coronal slices of the adult mouse brain assayed by two different single-cell resolution ST technologies, Xenium²⁰ and STARmap PLUS²¹ (Methods, Fig 6a-c).

For other tissues with substantially more inter-sample and inter-animal variation, alignment across serial sections is still achievable. For example, for serial sections of the developing human heart²², we can apply STalign to achieve structural correspondence (Methods, Fig 6d-f). Likewise, even for cancer tissues, which are highly non-prototypic in

structure, there is still often sufficient structural consistency across serial sections to enable alignment. As such, we have applied STalign to align single-cell resolution ST datasets arising from partially matched serial sections of the same breast cancer sample assayed by Xenium (Methods, Fig 6g-i). Likewise, we have applied STalign to align a single-cell ST dataset assayed by Xenium to a corresponding H&E image of the same tissue section (Methods, Fig 6j-l). We visually observe a high degree of spatial correspondence and overlap of structural features after alignment, highlighting STalign's applicability to diverse tissues.

Figure 6. Application of STalign to ST data of diverse tissues. **a.** Two coronal slices of the adult mouse brain assayed by two different single-cell resolution ST technologies, Xenium and STARmap PLUS **b.** Overlay of cellular positions before alignment. **c.** Overlay of cellular positions after alignment with STalign. **d.** Two single-cell resolution datasets from serial sections of the developing human heart. **e.** Overlay of cellular positions before

alignment. **f.** Overlay of cellular positions after alignment with STalign. **g.** Two single-cell resolution ST datasets from partially matched, serial breast cancer sections visualized as x- and y-coordinates of cellular positions. **h.** Overlay of cellular positions before alignment. **i.** Overlay of cellular positions after alignment with STalign. **j.** A single-cell resolution ST dataset with a corresponding H&E image from the same tissue section. **k.** Overlay of cellular positions and H&E image based on affine transformation by minimizing distances between manually placed landmarks, shown as points in red and turquoise. **l.** Overlay of cellular positions and H&E image after alignment with STalign.

7. In Fig. 2b: what are the genes for the three plots? One would guess they are the genes in Fig. 2a but there is no mention of this. Same for Fig. 3c.

We thank the reviewer for the request to clarify the genes to which the plots in Figure 2b and Figure 3c correspond. To improve clarity, we have updated the figure captions to include mention of the genes in Fig 2b and Fig 3c, provided below for the reviewer’s convenience.

Figure 2. Evaluation of STalign based on spatial gene expression correspondence. **a.** Correspondence of gene expression spatial organization between the target and aligned source for select spatially patterned genes. **b.** Transcript counts in the target compared to the aligned source at matched pixels for select genes: *Gabbr2*, *Gpr6*,

and *Cckar*. **c.** Distribution of cosine similarities between transcript counts in target versus aligned source at matched pixels for 457 spatially patterned genes with select genes marked. **d.** Spatial pattern of expression for a select non-spatially patterned gene in the target and aligned source. **e.** Counts for the target versus aligned source at matched pixels for select non-spatially patterned gene, *Mrgprf*. **f.** Distribution of cosine similarities between counts in target compared to the aligned source at matched pixels for 192 non-spatially patterned genes.

Figure 3. Application and evaluation of STalign on spatial transcriptomics data from different ST technologies based on normalized spatial gene expression correspondence. **A.** Overview of STalign on ST data from different ST technologies. Single-cell resolution ST is used as the source, with the initial image being produced from the x- and y-coordinates of each cell's position (top). For the multi-cellular resolution ST technologies, the corresponding single-cell resolution histological image is used as target (middle). STalign aligns the source to target (bottom). The manually placed landmarks that were utilized to improve alignment for these partially matched tissues are marked. **B.** Correspondence of gene expression spatial organization between the Visium target and aligned MERFISH source for select spatially patterned genes. **C.** Normalized gene expression in the Visium target compared to the aligned MERFISH source at matched spots and pseudospots respectively for select spatially patterned genes: *Baiap2*, *Slc17a6* and *Gpr151*. **D.** Distribution of cosine similarities between normalized gene expression in the Visium target versus aligned MERFISH source at matched spots and pseudospots for 227 spatially patterned genes detected by both ST technologies with select genes marked.

Reviewer #3 (Remarks to the Author):

The alignment of spatial transcriptomics (ST) data across slices, samples, and technologies is an important topic in the field. The authors have proposed STalign, a tool that aligns ST datasets by considering tissue section variations and non-linear distortions using diffeomorphic metric mapping. The authors have demonstrated that STalign achieves improved gene expression and cell-type correspondence compared to manual and landmark-based alignments.

Major concerns:

1. The authors have demonstrated that STalign significantly reduces the root mean square error (RMSE) between landmarks in single-cell ST datasets within technologies compared to alternative methods. To provide a more comprehensive analysis, it would be beneficial for the authors to showcase the gene expression correspondence across matched spatial locations for all the compared methods, not just STalign.

We thank the reviewer for their suggestions to strengthen the comparison of STalign with alternative methods beyond demonstrating that STalign reduces registration error between landmarks.

Based on the reviewer's suggestions to provide a more comprehensive analysis, we now showcase the gene expression correspondence across matched spatial locations for compared methods as well. Specifically, we have added Supp Fig 4b-c to the showcase examples of the gene expression correspondence across matched locations for the supervised affine transformation, provided below for the reviewer's convenience:

Supplemental Figure 4. Evaluation of STalign against supervised affine alignment. **a.** Spatial agreement of target and source that has been aligned based on a supervised affine transformation based on manually placed landmarks. **b.** Correspondence of gene expression spatial organization between the target and supervised affine aligned source for select spatially patterned genes. **c.** Transcript counts in the target compared to the supervised affine aligned source at matched pixels for select genes: *Cckar*, *Htr5b*, *Gabbr2* and *Ackr2*. **d.** Cosine similarities between transcript counts in target versus aligned source for STalign compared to affine alignment for 457 spatially patterned genes. (mean difference = 0.09) Genes featured in Supplemental Figure 4b-c, Figure 2a-c, and Supplemental Figure 2 are highlighted. **e.** Cosine similarities between transcript counts in target versus aligned source for STalign compared to manual alignment for 192 non-spatially patterned genes. (mean difference = 0.02) Genes featured in Figure 2d-f and Supplemental Figure 3 are highlighted.

2. The alignment performed by STalign relies on tissue structures formed by cell densities. In the case of imaging-based ST, cell segmentation is typically utilized to identify cells, introducing additional uncertainties. It would be important for the authors to discuss whether this step has any influence on the performance of STalign and address its potential impact.

We thank the reviewer for bringing up this important point. Indeed, alignments of imaging-based ST data will be dependent on the quality of the cell segmentation and the extent to which these cell segmentations are representative cell densities that may reveal tissues structures. In theory, a failed cell segmentation or cell segmentations that rely on cell-type-specific stains or RNAs would render the derived cell density to be no longer be representative of the profiled tissue structure and would therefore be very challenging to align based on cell density. We have expanded upon these caveats in the discussion, included below for the reviewer's reference.

(line 595) STalign further relies on the representative nature of cell segmentations in ST data to reflect underlying tissue structures. As such, limitations in cell segmentations that render the derived cell density to be no longer be representative of the profiled tissue structure could present challenges for alignment with STalign.

In practice, we have been able to align ST datasets processed using very different cell segmentation stains and algorithms. To further this point, we have now applied STalign to align coronal sections of the mouse brain profiled by Xenium where cell segmentation was achieved using a watershedding algorithm on DAPI staining images and STARmap where cell segmentation was achieved using a Random Forest classifier based approach on fluorescent Nissl staining images (Figure 6a-c), provided below for the reviewer's reference:

Figure 6. Application of STalign to ST data of diverse tissues. **a.** Two coronal slices of the adult mouse brain assayed by two different single-cell resolution ST technologies, Xenium and STARmap PLUS **b.** Overlay of cellular positions before alignment. **c.** Overlay of cellular positions after alignment with STalign. **d.** Two single-cell resolution datasets from serial sections of the developing human heart. **e.** Overlay of cellular positions before

alignment. **f.** Overlay of cellular positions after alignment with STalign. **g.** Two single-cell resolution ST datasets from partially matched, serial breast cancer sections visualized as x- and y-coordinates of cellular positions. **h.** Overlay of cellular positions before alignment. **i.** Overlay of cellular positions after alignment with STalign. **j.** A single-cell resolution ST dataset with a corresponding H&E image from the same tissue section. **k.** Overlay of cellular positions and H&E image based on affine transformation by minimizing distances between manually placed landmarks, shown as points in red and turquoise. **l.** Overlay of cellular positions and H&E image after alignment with STalign.

3. As discussed by the authors, achieving a one-to-one correspondence between cells across samples is challenging, particularly for pathological tissues. In such scenarios, if landmark information is available, it would be interesting to explore whether STalign can incorporate known tissue structures or features to enhance alignment. Currently, the authors only explore these features to generate initial values, but further exploration or discussion could be beneficial.

We thank the reviewer for the opportunity to clarify this important point. Indeed, as the reviewer notes, STalign allows users to incorporate landmark information to generate an initial affine transformation. However, STalign also incorporates the option to include these landmarks in our objective function. We have now clarified this point in the Online Methods (section 1.5 Image registration), provided below for the reviewer's reference:

For improved robustness, our software allows users to input pairs of corresponding points in the source and target images. These points can be used either to initialize the affine transformation A through least squares (steering our gradient based solution toward an appropriate local minima in this challenging nonconvex optimization problem); or can be used to drive the optimization problem itself by modifying to our objective function to be $E(A, v) = R(v) + M_{\theta}(\phi^{A, v} \cdot I^S, I^T) + P(\phi^{A, v}(X^S), X^T)$ such that

$$P(\phi^{A, v}(X^S), X^T) = \frac{1}{2\sigma_P^2} \sum_{i=1}^N |\phi^{A, v}(X_i^S) - X_i^T|^2$$

where X_i^S and X_i^T are the i th point of N corresponding points in the source and target respectively and σ_P^2 is a user tunable parameter that adjusts balance between matching corresponding landmark points, matching images, and regularization, where large values correspond to less accuracy matching points and small values correspond to more accuracy matching points. Landmark based optimization in the LDDMM framework has been studied extensively (see for example [9]).

We have further revised the main text to emphasize the availability of this option:

(Line 218) Specifically, we applied STalign to align the previously analyzed single-cell resolution ST dataset of a full coronal slice of the adult mouse brain assayed by MERFISH to a multi-cellular pixel resolution ST dataset of an analogous hemi-brain slice assayed by Visium (Fig 3a). As such, in addition to being from different ST technologies, these two ST datasets further represent

partially matched tissue sections. Because of this partial matching, we incorporated manually placed landmarks to initialize the alignment as well as further help steer our gradient descent towards an appropriate solution (Online Methods).

4. After aligning spatial transcriptomics (ST) data across different platforms, we obtain gene expressions for matched spatial locations. However, it is important to consider that different ST platforms may have varying gene coverage, and batch effects can also be present. In light of these considerations, what types of downstream analyses could be conducted based on these matched spatial observations?

We thank the reviewer for the opportunity to clarify this important point. Indeed, when comparing gene expressions for matched spatial locations, particularly in a health versus disease setting, it will be important to control for platform-specific effects due to their varying gene coverage and other batch effects in order to minimize confounders. In light of these considerations, we would not recommend confounding spatial comparisons across case and control settings with different spatial transcriptomics platforms for example.

However, aligning spatial transcriptomics data across different platforms may still be desirable for a number of downstream analyses. For example, we may be interested in leveraging the unique strengths of different spatial transcriptomics platforms by applying them to serial sections from the same tissue block. Given that differential spatial transcriptomics platforms currently prioritize either resolution or genome-wide capabilities, we may wish to apply a spot-resolution full-transcriptome ST technology on one serial section, and then a targeted molecular resolution ST technology on the next serial section, and use spatial alignment to effectively obtain both types of measurements at the same matched spatial location.

Likewise, aligning spatial transcriptomics data across different platforms may allow us to better interrogate platform-specific differences and strengths. For example, we may be interested in understanding which spatial transcriptomics technology has better sensitivity for transcription factors within the distal CA1 region of the hippocampus. Aligning spatial transcriptomics data across different technologies could allow for this evaluation to identify the appropriate matched spatial location.

Finally, as we have shown, aligning these different ST datasets to a CCF such as the Allen Brain Atlas, enable automated lift over a common set of atlas structural annotations. Even though different ST platforms may have varying gene coverage or batch effects, we would anticipate that the biological insights they reveal regarding these annotated structures to be consistent. For example, if one ST technology suggests that the gene *Grm2* is highly expressed in the dentate gyrus annotated structure, I would expect other ST technologies to also affirm this finding provided the *Grm2* is measured by those technologies as well. Aligning spatial transcriptomics data across different technologies to the same CCF could therefore facilitate this type of standardization and unification of biological insights.

We have incorporated these important points in the revised discussion, included below for the reviewer's reference.

(line 526) We anticipate that future applications of STalign to ST data particularly across ST technologies will enable cross-technology comparisons as well as cross-technology integration through spatial alignment. In particular, aligning ST data for similar tissues across different ST technology platforms may allow us to better interrogate platform-specific differences and strengths. Likewise, given that different ST technologies currently generally prioritize either resolution or genome-wide capabilities, we may wish to apply different ST technologies on serial sections to leverage their unique strengths to characterize matched spatial location. With atlasing efforts like The Human BioMolecular Atlas Program and others producing 3D CCFs²⁴, application of STalign to align ST data to such CCFs to enable automated lift over of atlas structural annotations will further facilitate standardization and unification of biological insights regarding annotated structures.

5. The authors have made a Python package with nice tutorials for various usage examples. However, it would be helpful to include a tutorial specifically addressing single-cell and spot-resolution ST alignment. Additionally, providing instructions on preparing a single-cell resolution HE staining image obtained from Visium would greatly assist researchers using this technique.

We thank the reviewer for their appreciation of our tutorials with usage examples and we agree that it would be helpful to include a tutorial specifically addressing single-cell and spot-resolution ST alignment. As recommended, we have added the notebook, "Aligning single-cell resolution spatial transcriptomics data to H&E staining image from Visium" at <https://jef.works/STalign/notebooks/merfish-visium-alignment-with-point-annotator.html>.

We demonstrate in this notebook that the only two preprocessing steps done to the H&E staining image obtained from Visium prior to alignment is normalizing using STalign.normalize() in case there are any outlier intensities and transposing the image from a NxMx3 matrix to a 3xNxM matrix.

Minor concerns:

1. Line 183: The authors mention a "matching probability > 0.85." It would be helpful if the authors could provide clarification on how this quantity is calculated or derived.

We agree with the reviewer that it will be helpful to provide more clarification regarding the matching probabilities. In order to align partially matched tissue sections, we use a Gaussian mixture model. We have now updated the Online Methods (section 1.5 Image registration) with more details regarding this Gaussian mixture model:

Briefly, we use 3 classes in our Gaussian mixture model (background, artifact, and pixels to be matched). Each is modeled as a Gaussian random variable with a unknown mean (optionally the mean can be assumed known and specified as an input parameter), a known variance (specified as an input parameter), and an unknown prior probability. While the means for background and artifact are constant values, the mean for pixels to be matched is equal to $f_{\theta}[\phi^{A,v} \cdot I^S](x)$ and is a function of space. Parameters are estimated by standard techniques, and $W(x)$ is computed as the posterior probability that the pixel at x belongs to the "pixels to be matched" class.

Therefore, by "matching probability > 0.85", we mean the points at pixels that have a posterior probability of belonging to the "pixels to be matched" class greater than 85%, with the 85% threshold being chosen manually based on visual inspection. We have also clarified this in the main methods:

(Line 1024) Given that the MERFISH tissue section is larger than the Visium, we considered the aligned region to be limited to the MERFISH tissue that had a matching probability > 0.85 based on the posterior probability of pixels belonging to the matched class in the Gaussian mixture modeling, with the 0.85 threshold being manually chosen based on visual inspection. We restricted the set of cells in the MERFISH dataset to only those in this aligned region for downstream evaluation.

We have further updated STalign.py to make it easier for users to use this matching probability output and further updated the tutorial notebook, "Aligning single-cell resolution spatial transcriptomics data to H&E staining image from Visium" with an example of how to use this matching probability with relevant screenshots below for the reviewer's reference:

Stalign

Search docs

Stalign

Tutorials

- Aligning two coronal sections of adult mouse brain from MERFISH
- Aligning partially matched, serial, single-cell resolution breast cancer spatial transcriptomics data from Xenium
- Aligning single-cell resolution breast cancer spatial transcriptomics data to corresponding H&E staining image from Xenium
- Aligning partially matched coronal sections of adult mouse brain from Xenium and STARmap PLUS
- Aligning adult mouse coronal brain sections with the Allen Brain Atlas
- Aligning partial coronal brain sections with the Allen Brain Atlas
- Aligning single cell resolution spatial transcriptomics data to H&E staining image from Visium
- Aligning heart ST data from ISS

Functions

```
[56]: # compute weight values for transformed source points from target image pixel locations on
      # tasM = Stalign.interp(X1,X2,W[None],floats1,targets[None],permut=1,8,1, float())
      fig,ax = plt.subplots()
      scatter = ax.scatter(targets[:,1],targets[:,0],c=targets[:,0],s=0.1,marker='o',label='1')
      legend = ax.legend(scatter.legend_elements(),
                        loc='upper right', title='W values')
      ax.invert_yaxis()
```

```
[57]: # save weight values
      results['W_values'] = tasM(W, R)
```

```
[58]: results
```

```
[58]: results
```

Unnamed: 0	row	volume	center_x	center_y
0	15838042624264719696434349950479	33	532.778772	617.916619
1	240594727341603723397640542692813003	33	1054.43016	594.808018
2	3076430460700812309199503248604719910662	33	1267.483208	576.880018
3	3086330346076316429997331474071348973	33	1403.401822	572.616017
4	31316271818409762168867924337202142401	33	507.949497	608.364018
...
8963	3117040240348913398616843678992293	1345	1621.490809	9423.123179
8964	3128188005098327776181237820209399759	1345	901.023435	9425.047379
8965	3322999186990281339501510609798812698	1345	439.647325	9405.470978
8966	130462787596702108458136479486023681	1346	778.366670	9406.874978
8967	248870416707441542727758918474700721	1346	592.088115	9407.841178

85958 rows x 5 columns

```
[59]: fig,ax = plt.subplots()
      ax.hist(results['W_values'], bins = 20)
```

```
[59]: array([41263., 819., 538., 438., 374., 361., 361., 256.,
        202., 276., 154., 129., 146., 198., 134., 235.,
        295., 356., 787., 30766.])
array([0., 0.04999861, 0.04999861, 0.14999872, 0.19999863,
        0.24999854, 0.29999845, 0.34999837, 0.39999828, 0.44999819,
        0.4999981, 0.54999802, 0.59999793, 0.64999784, 0.69999775,
        0.74999766, 0.79999757, 0.84999748, 0.89999739, 0.9499973,
        0.99999721])
<BarContainer object of 20 artists>
```

Based on this distribution and manual inspection, we can manually set a threshold value to filter transformed MERFISH data.

```
[61]: # set a threshold value
      Wthresh = .95

      # filter transformed MERFISH data
      results_filtered = results[results['W_values'] > Wthresh]

      # plot
      fig,ax = plt.subplots()
      ax.scatter(results_filtered['aligned_x'],results_filtered['aligned_y'],c=results_filtered['W_values'])
      ax.set_aspect('equal')
      ax.invert_yaxis()
      plt.show()
```

2. It would also be beneficial if the authors could include information regarding the computational times taken for STalign, as this would provide insights into the tool's efficiency and practicality.

We thank the reviewer for this important suggestion. We agree it will be beneficial to include information regarding the computational times for STalign. However, it is important to note that STalign's runtime will vary depending on the users' system setup, particularly between GPU and CPU settings.

Still, to provide users with guidance on STalign's efficiency and practicality, we have now provided an evaluation of the runtime for the STalign.LDDMM function, the most computationally intensive component, on various alignments in our Jupyter notebook tutorials. In general, we have found runtimes ranging from less than a minute to a few hours depending on the size of the dataset, the number of epochs in the alignment, and other system variables, as described in the table below and incorporated into our revision. Likewise, we note that runtimes can be substantially reduced with GPU settings. For example, when aligning a MERFISH dataset and Visium H&E image with LDDMM after an initial affine alignment using a few landmark points, the GPU and CPU settings completed the LDDMM function with the same exact parameters in approximately 5 minutes and 35 minutes, respectively. We have incorporated these results in a supplementary table and in the main methods, included below for the reviewer's reference.

(Methods)

Runtime Estimate. Runtime for the STalign.LDDMM function was estimated for CPU settings using a MacBook Pro with an 2.4 GHz 8-Core Intel Core i9 processor and 32 GB 2400 MHz DDR4 memory, and for GPU settings using an Intel Xeon W-3365 2.7GHz Thirty-Two Core 48MB 270W processor with 8 x DDR4-3200 16GB ECC Reg memory and a NVIDIA RTX A5000 24GB PCI-E video card.

Jupyter notebook	STalign.LDDMM niter	GPU runtime	CPU runtime
merfish-visium-alignment-with-point-annotator.ipynb	200	GPU times: user 5min 6s, sys: 22.8 s, total: 5min 29s Wall time: 5min 3s	CPU times: user 35min 57s, sys: 2min 43s, total: 38min 41s Wall time: 9min 50s
merfish-merfish-alignment.ipynb	10000	GPU times: 32min 43s, sys: 10.9s, total: 32min 54s	CPU times: user 2h 39min 42s, sys: 31min 2s, total: 3h 10min 45s

		Wall time: 22min 47s	Wall time: 1h 27min 7s
xenium-xenium-alignment.ipynb	200		CPU times: user 43.2 s, sys: 5.2 s, total: 48.4 s Wall time: 41.1 s
xenium-heimage-alignment.ipynb	2000		CPU times: user 3min 22s, sys: 24.2 s, total: 3min 47s Wall time: 3min 16s
xenium-starmap-alignment.ipynb	4000		CPU times: user 13min 36s, sys: 1min 52s, total: 15min 28s Wall time: 14min
merfish-allen3Datlas-alignment.ipynb	2000		CPU times: user 3h 24min 29s, sys: 21min 43s, total: 3h 46min 13s Wall time: 1h 27min 45s
starmap-allen3Datlas-alignment.ipynb	800		CPU times: user 2h 36min 26s, sys: 1h 16min 5s, total: 3h 52min 32s Wall time: 13min 27s
heart-alignment.ipynb	1000		CPU times: user 33min 13s, sys: 1min 56s, total: 35min 10s Wall time: 3min 53s

Supplementary Table 2. Runtime estimates for the STalign.LDDMM and the STalign.LDDMM_3D_to_slice functions for different ST data alignments.

With regards to future practicality, the performance of each iteration of STalign.LDDMM scales with respect to the number of pixels in the rasterized image. When generating the rasterized image the user is able to select the pixel resolution and therefore has control over the number of pixels. Solving the diffeomorphic alignment transformation through calculating the Fourier transform has $O(n \log n)$ performance for each iteration of the gradient descent where n is the number of pixels. However, the slowest process in STalign.LDDMM is the interpolation required to generate the transformed source image with $O(n)$ performance. We have incorporated commentary on the alignment algorithm performance and practicality in the revised discussion, included below for the reviewer's reference.

DISCUSSION

(line 518) As ST technologies continue to evolve, we anticipate STalign will continue to be applicable due to our use of rasterization to convert the positions of single cells into an image with specified resolution. The runtime of each iteration of the STalign alignment algorithm scales with respect to the number of pixels in this image. For most evaluated datasets, we find that STalign is generally able to converge onto an optimal alignment within a few minutes to a few hours, depending on the number of pixels, the number of iterations, and other system variables (Methods, Supp Table 2). Whereas other alignment algorithms generally scale in memory and runtime with the number of spatially resolved measurements (spots or cells)^{1,2}, which will likely make them computationally untenable as ST technologies evolve to increase the number of spatially resolved measurements that can be assayed. Overall, we anticipate that the ability for users to choose the rasterization resolution, and therefore the number of pixels in the rasterized image, will allow STalign to maintain its utility for larger datasets.

Reviewer #1 (Remarks to the Author):

With the added results and explanations, most of my concerns have been addressed in the updated manuscript. The method is novel and the alignment performance is convincing. I only have the following clarification questions.

1. Fig 6 is to demonstrate that STalign can apply to multiple technologies, but it's not clear for some panels what technologies they are showing. Can you specify the technology for each panel in the main text and figure legend?
2. STalign is flexible to handle only cell density or incorporate certain features (e.g. RGB channels in images), so what is the exact input to STalign for H&E images and CCF? Specifically, is cell segmentation performed on H&E images and only cell density is used, or are pixel-wise RGB values used? CCF has a 50 μ m resolution, so it doesn't have cell densities, what is the feature space of CCF that the f_{θ} polynomial function fits to?
3. In the 2D to 3D alignment, the alignment seems to fall onto a 2D hyperplane in 3D space in the new plot. Is that true? Or is there quantitative measurement to assess how far the alignment goes out of a 2D hyperplane?

Reviewer #2 (Remarks to the Author):

The reviewer would like to thank the authors for their efforts to address the reviewer's comments. The reviewer has further suggestions on two of the points raised previously.

1. Regarding previous comment "The paper mentioned that differences between different samples can come from both biological variations and technical variations, as well as distortions between tissues. As these have not been discussed much before, it would be helpful if the authors can expand the discussion on this, which can also make the goal of stAlign more clear. For example, where can technical variations come from? Is stAlign designed to remove the technical variations and retain biological variations, and why?": it's great that the authors clarified different sources of variations between samples. However, there is no clear answer as to what are the preferred scenarios stAlign can be applied to – for example, samples from different experimental platforms and the same condition, samples from the same platforms and different conditions. From the authors' responses to Reviewer 3's point 4, it seems that "samples from different experimental platforms and the same condition" is an appropriate application scenario and "samples from different experimental platforms and different conditions" is a challenging scenario. It would help users to decide if this is the right tool to use if this point is made clear.
2. Regarding previous comment "This is a general and crucial point for all the alignment results: so far the results focus on showing that they alignment is performed properly. However, one major goal of aligning samples is to find differences between samples. Such differences have not been discussed much in the results in Lines 142-146, much of it was attributed to noise. To show the application of stAlign in helping to compare samples, it is suggested to give one or two examples of meaningful differences found between samples, such as genes that have different spatial patterns or different cell types distributions in space. ": Although the authors provided potential use cases of applying stAlign to find differences between samples, it would strengthen the results if these analyses are actually performed and results are included in this manuscript.

Reviewer #3 (Remarks to the Author):

The authors have addressed all my previous concerns. I am happy with the revised version of the manuscript and would like to recommend that it is accepted for publication.

Point by Point Response – Overview

We sincerely thank the editor and the reviewers for their insightful and constructive feedback in helping us improve this manuscript. We have now revised the manuscript to address all the points raised by the reviewers, organized herein as a point-by-point response. Throughout this point-by-point response, reviewer comments are shown in **blue**, with our responses in **green**, and changes to the manuscript in **black**.

Reviewer #1 (Remarks to the Author):

With the added results and explanations, most of my concerns have been addressed in the updated manuscript. The method is novel and the alignment performance is convincing. I only have the following clarification questions.

We thank the reviewer for their feedback and assistance in clarifying the details of our manuscript for a broader audience.

1. Fig 6 is to demonstrate that STalign can apply to multiple technologies, but it's not clear for some panels what technologies they are showing. Can you specify the technology for each panel in the main text and figure legend?

The reviewer is correct that Fig 6 now demonstrates that STalign can be applied to multiple technologies. We have now clarified the specific technologies that each panel is demonstrating.

Specifically, for Fig 6d, the serial sections of developing human heart were assayed with *in situ* sequencing as described in "Asp, M. *et al.* A Spatiotemporal Organ-Wide Gene Expression and Cell Atlas of the Developing Human Heart. *Cell* **179**, 1647-1660.e19 (2019)". For Fig 6g and Fig6j, the human breast cancer datasets Replicate 1 and Replicate 2 were assayed by Xenium *in situ* sequencing. Fig 6j also includes a H&E image resulting from staining Replicate 1 after Xenium processing.

We have now provided these clarifications in the main text as well as updated the figure and captions with relevant excerpts below for the reviewer's convenience:

Figure 6

Figure 6. Application of STalign to ST data of diverse tissues. **a.** Two coronal slices of the adult mouse brain assayed by two different single-cell resolution ST technologies, Xenium and STARmap PLUS **b.** Overlay of cellular positions before alignment. **c.** Overlay of cellular positions after alignment with STalign. **d.** Two single-cell resolution datasets, **Replicate 1 and Replicate 2**, from serial sections of the developing human heart **assayed by ISS** and visualized as x- and y-coordinates of cellular positions. **e.** Overlay of cellular positions before alignment. **f.** Overlay of cellular positions after alignment with STalign. **g.** Two single-cell resolution ST datasets, **Replicate 1 and Replicate 2**, from partially matched, serial breast cancer sections assayed by Xenium and visualized as x- and y-coordinates of cellular positions. **h.** Overlay of cellular positions before alignment. **i.** Overlay of cellular positions after alignment with STalign. **j.** **Replicate 1 from Xenium single-resolution breast cancer dataset** with a corresponding H&E image from the same tissue section. **k.** Overlay of cellular positions and H&E image based on affine transformation by minimizing distances between manually placed landmarks, shown as points in red and turquoise. **l.** Overlay of cellular positions and H&E image after alignment with STalign.

STalign applicable to diverse tissues profiled by diverse ST technologies

...

For other tissues with substantially more inter-sample and inter-animal variation, alignment across serial sections is still achievable. For example, for serial sections of the developing human **heart assayed at single-cell resolution with *in situ* sequencing (ISS)**²³, we can apply STalign to achieve structural correspondence (Methods, Fig 6d-f).

2. STalign is flexible to handle only cell density or incorporate certain features (e.g. RGB channels in images), so what is the exact input to STalign for H&E images and CCF? Specifically, is cell segmentation performed on H&E images and only cell density is used, or are pixel-wise RGB values used? CCF has a 50 μ m resolution, so it doesn't have cell densities, what is the feature space of CCF that the f_{θ} polynomial function fits to?

We thank the reviewer for the opportunity to clarify the inputs to STalign for the H&E images and the CCF.

For ST datasets for which H&E staining has been performed on the sample, the H&E data is provided as an RGB image. We use RGB values at each pixel directly as the input. We do not perform cell segmentation on the H&E data. Pixel values in a RGB H&E image are reflective of the layout of cells such that these images have a contrast profile similar enough to be aligned with cell density images.

The CCF we use for alignment is the 3D 50 μ m resolution grayscale "ara_nissl" dataset (Fig5a) which can be acquired from Allen Institute API at <https://help.brain-map.org/display/mouseconnectivity/API> or downloaded directly at the following link: http://download.alleninstitute.org/informatics-archive/current-release/mouse_ccf/ara_nissl/ara_nissl_50.nrrd. These Nissl images do not strictly show cells, but because Nissl stains Nissl bodies in neurons, the intensity of the voxels is reflective of neuronal density even at the 50 μ m voxel resolution that we utilized for alignment. As such, the images have a contrast profile that is similar enough that our polynomial transforms can be used to estimate cell density sufficiently well for registration. Note that the Nissl image was an input to our function f , and the output of the function f is a rescaled image whose contrast profile is a sufficiently good estimator of cell density.

We have updated the main text and methods to reflect how we use the H&E images and the format of the CCF. Excerpts are included below:

Results

Overview of Method

To align two ST datasets, STalign solves a mapping that minimizes the dissimilarity between a source and a target ST dataset subject to regularization penalties (Online Methods). Within single-cell resolution ST technologies, both the source and target ST datasets are represented as cellular positions (x^{ρ_s}, y^{ρ_s}) and (x^{ρ_t}, y^{ρ_t}) respectively (Fig 1a). Solving the mapping with respect to single cells has quadratic complexity and is computationally intractable, so STalign applies a rasterization approach to reduce computational time (Fig 1b). Briefly, STalign models the positions of single cells as a marginal space measure ρ within the varifold measure framework⁶. STalign then convolves the space measure ρ with Gaussian kernels k to obtain the smooth, rasterized function $I(x, y) = \left[k^{\frac{1}{2}} * \rho \right] (x, y)$. Finally, STalign samples from the continuous $I(x, y)$ to get a discrete image of a specified size with a specified pixel resolution. STalign focuses on minimizing the dissimilarity between the source and target images I^S and I^T rather than minimizing the dissimilarity between the source and target space measures because, while approximately equivalent, the former can be calculated more efficiently (Online Methods). **For ST technologies or atlases that are not single-cell resolution, instead of rasterizing cellular positions to generate an image, we can utilize other standard image types for alignment including red-green-blue images of hematoxylin and eosin (H&E) stained tissue.** To solve for a mapping that minimizes the dissimilarity between source and target images I^S and I^T , STalign utilizes the LDDMM framework (Fig 1c).

...

STalign enables alignment of ST datasets across technologies

Many technologies for spatially resolved transcriptomic profiling are available, varying in experimental throughput and spatial resolution⁹. We thus applied STalign to align two ST datasets from two such different ST technologies. Specifically, we applied STalign to align the previously analyzed single-cell resolution ST dataset of a full coronal slice of the adult mouse brain assayed by MERFISH to a multi-cellular pixel resolution ST dataset of an analogous hemi-brain slice assayed by Visium (Fig 3a). As such, in addition to being from different ST technologies, these two ST datasets further represent partially matched tissue sections. Because of this partial matching, we incorporated manually placed landmarks to initialize the alignment as well as further help steer our gradient descent towards an appropriate solution (Online Methods). **For the MERFISH data, we rasterized the cell positions to generate the source image. For the Visium dataset, the target image for alignment was a red-green-blue single-cell resolution H&E staining image obtained from the same tissue section as the gene expression data (Methods).**

...

STalign enables alignment of ST datasets to a 3D common coordinate framework

...

We thus applied STalign to align ST datasets to a 3D common coordinate framework (CCF). Specifically, we applied STalign to align 9 ST datasets of the adult mouse brain assayed by

MERFISH to a **3D** 50µm resolution **greyscale volume of the** adult mouse brain established by the Allen Brain Atlas¹¹ (Methods, Fig 5a).

Methods

Datasets

...

The CCF and brain region annotations were obtained from the Allen Brain Atlas API <https://help.brain-map.org/display/mouseconnectivity/API> . The 50µm resolution 3D adult mouse brain CCF **used for alignment** was ara_nissl_50 downloaded directly at (http://download.alleninstitute.org/informatics-archive/current-release/mouse_ccf/ara_nissl/ara_nissl_50.nrrd). The brain region annotations that correspond to the 50µm resolution CCF were ccf_2017/annotation_50 downloaded directly at (https://download.alleninstitute.org/informatics-archive/current-release/mouse_ccf/annotation/ccf_2017/annotation_50.nrrd).

Application of STalign

...

To align a MERFISH dataset to a Visium dataset, we applied STalign with MERFISH Slice 2 Replicate 3, rasterized at a 50µm resolution, as the source and the **red-green-blue** high resolution Visium hematoxylin and eosin (H&E) staining image as the target.

...

To align MERFISH to the Allen CCF, we applied STalign using the 3D reconstructed Nissl image from the Allen CCF atlas as a source, and each of our 9 MERFISH images as a target. **The Nissl CCF is a 50µm resolution grayscale volume that was constructed by registering 2D Nissl images to a 3D anatomical template. The template this Nissl dataset was registered to was constructed by averaging high-resolution customized serial two-photon tomography scans of 1675 brains.**

...

To align Xenium to H&E, we applied STalign with Xenium Breast Cancer Replicate 1, rasterized at 30µm resolution, as the target and the corresponding **red-green-blue** H&E image from the same tissue as the source.

3. In the 2D to 3D alignment, the alignment seems to fall onto a 2D hyperplane in 3D space in the new plot. Is that true? Or is there quantitative measurement to assess how far the alignment goes out of a 2D hyperplane?

We thank the reviewer for the opportunity to better explain the alignment of the 2D section with the 3D atlas. It is not true that the alignment falls onto a 2D plane in 3D space. There is a

curvature of the aligned slice which we have now quantified by calculating the root mean square error of the aligned slice with respect to best-fit 2D plane. We have updated the main text to emphasize this point with relevant excerpts provided below:

STalign enables alignment of ST datasets to a 3D common coordinate framework

...

Specifically, we applied STalign to align 9 ST datasets of the adult mouse brain assayed by MERFISH to a 3D 50 μ m resolution greyscale volume of the adult mouse brain established by the Allen Brain Atlas¹¹ (Methods, Fig 5a). **We note that when aligning 2D ST datasets to a 3D CCF, there may not exist a perfect 2D hyperplane in the 3D space in which the alignment falls. For example, when MERFISH Slice 2 Replicate 2 was aligned to the Allen Brain Atlas, the RMSE with respect to the best-fit 2D plane is 1.255 μ m (Methods, Fig 5b); this non-zero RMSE highlights that the slice has deformities that extend out of the 2D hyperplane. Additionally, the angle of the best-fit 2D plane with respect to the y-z plane of the Allen Brain Atlas is 9.07 degrees; this non-zero angle demonstrates that the slice does not align with a strictly coronal section. As such, our 2D-3D alignment accommodates global deformations (i.e. 2D plane with respect to the 3D atlas) as well as local distortions (i.e. deformations in and out of a 2D plane) using the same underlying LDDMM framework as 2D alignments (Online Methods).**

Methods:

Quantification of local and global distortions

To evaluate the local deformation of the 2D MERFISH Slice 2 Replicate 2 when aligned to the 3D Allen Brain Atlas CCF, the STalign function LDDMM_to_slice() was run with the following parameters: nt=4, niter=2000, sigmaA = 2, sigmaB = 2, sigmaM = 2, muA = [3,3,3], muB = [0,0,0]. Then, the least squares fit method was used to evaluate the most representative plane for the transformed slice, and the root mean square error was calculated. The angle between the most representative plane and the y-z plane, which runs through the medio-lateral and dorso-ventral axis of the Allen Brain Atlas, was computed.

Reviewer #2 (Remarks to the Author):

The reviewer would like to thank the authors for their efforts to address the reviewer's comments. The reviewer has further suggestions on two of the points raised previously.

1. Regarding previous comment "The paper mentioned that differences between different samples can come from both biological variations and technical variations, as well as distortions between tissues. As these have not been discussed much before, it would be helpful if the authors can expand the discussion on this, which can also make the goal of stAlign more clear. For example, where can technical variations come from? Is stAlign designed to remove the technical variations and retain biological variations, and why?": it's great that the authors

clarified different sources of variations between samples. However, there is no clear answer as to what are the preferred scenarios stAlign can be applied to – for example, samples from different experimental platforms and the same condition, samples from the same platforms and different conditions. From the authors' responses to Reviewer 3's point 4, it seems that "samples from different experimental platforms and the same condition" is an appropriate application scenario and "samples from different experimental platforms and different conditions" is a challenging scenario. It would help users to decide if this is the right tool to use if this point is made clear.

We thank the reviewer for the opportunity to clarify how we anticipate STalign should be applied. We have now updated the discussion to specifically clarify that STalign should be applied to samples assayed by different ST technologies and the same condition as well as samples assayed by the same ST technologies in a different conditions as suggested by the reviewer.

Discussion

...

We anticipate that future applications of STalign to ST data **from serial sections or conditionally matched tissues** across **different** ST technologies will enable cross-technology comparisons as well as cross-technology integration through spatial alignment. In particular, **applying STalign to align ST data of structurally similar tissues from the same condition** across different ST technology platforms may allow us to better interrogate platform-specific differences and strengths. Given that different ST technologies currently generally prioritize either resolution or genome-wide capabilities, **applying STalign to align serial sections assayed by different ST technologies may allow us to leverage each technology's unique strengths to characterize matched spatial location.**

...

We further anticipate future applications of STalign to ST data from structurally matched tissues in case-control settings **assayed by the same ST technologies** will enhance the throughput for yielding meaningful comparisons regarding gene expression and cell-type distributions in space as evidenced by recent applications of ST technologies to characterize spatially-resolved age-related²⁷ and injury-related²⁸ gene expression variation.

2. Regarding previous comment "This is a general and crucial point for all the alignment results: so far the results focus on showing that they alignment is performed properly. However, one major goal of aligning samples is to find differences between samples. Such differences have not been discussed much in the results in Lines 142-146, much of it was attributed to noise. To show the application of stAlign in helping to compare samples, it is suggested to give one or two examples of meaningful differences found between samples, such as genes that have different spatial patterns or different cell types distributions in space. ": Although the authors provided potential use cases of applying stAlign to find differences between samples, it would strengthen the results if these analyses are actually performed and results are included in this

manuscript.

We thank the reviewer for their suggestion and agree that it would be interesting to perform STalign to find biologically meaningful spatial gene expression differences between samples in a healthy vs disease setting. Indeed, making new biologically meaningful discoveries is an exciting prospective for the application of STalign.

However, we caution that in order to make statistically rigorous assessments of such differences, there is a need for additional methodological development that takes into consideration multiple testing across spatial locations and across genes, and other technical challenges. As the reviewer notes, we have highlighted previously published research papers where spatial gene expression differences were observed between healthy vs disease tissues and described but not statistically evaluated likely for this reason.

Developing such statistically rigorous assessments of biologically meaningful differences is beyond the scope of this current manuscript. Likewise, while we believe it was appropriate for previous papers to describe spatial differences without statistical evaluation given that these differences were validated by other experiments and contextualized in biologically-focused stories, it would be inappropriate to simply describe spatial differences without statistical evaluation given the mathematical nature of this manuscript.

While we and others are currently applying STalign to identify spatial gene expression differences between healthy vs disease tissues, we believe it will be important to provide the appropriate interpretations and further validations of these differences using orthogonal assays such as immunohistochemistry in order to ensure their biological validity prior to publication. Such interpretations and validations are beyond the scope of this manuscript, which ultimately focuses on computational method development. Given the limitations on manuscript length and the other points noted previously, we believe it is more appropriate to address this aspect in future, dedicated follow-up studies.

Overall, our current paper primarily focuses on demonstrating the utility and effectiveness of our computational method, serving as a foundational step. We believe these examples are sufficient for readers to be able to appreciate the potential applications of STalign. We will look forward to sharing results applying STalign to find biological meaningful differences between healthy vs disease tissues with the scientific community in the future.

Reviewer #3 (Remarks to the Author):

The authors have addressed all my previous concerns. I am happy with the revised version of the manuscript and would like to recommend that it is accepted for publication.

Reviewer #1 (Remarks to the Author):

The revised manuscript addressed all my previous concerns and confusions. I would like to recommend it for publication.